# Biotic and abiotic controls on co-occurring nitrogen cycling processes in shallow Arctic shelf sediments

N.D. McTigue[1,†], W.S. Gardner[1], K.H. Dunton[1] & A.K. Hardison[1]

The processes that convert bioavailable inorganic nitrogen to inert nitrogen gas are prominent in continental shelf sediments and represent a critical global sink, yet little is known of these pathways in the Arctic where 18% of the world's continental shelves are located. Moreover, few data from the Arctic exist that separate loss processes like denitrification and anaerobic ammonium oxidation (anammox) from recycling pathways like dissimilatory nitrate reduction to ammonium (DNRA) or source pathways like nitrogen fixation. Here we present measurements of these co-occurring processes using $^{15}$N tracers. Denitrification was heterogeneous among stations and an order of magnitude greater than anammox and DNRA, while nitrogen fixation was undetectable. No abiotic factors correlated with interstation variability in biogeochemical rates; however, bioturbation potential explained most of the variation. Fauna-enhanced denitrification is a potentially important but overlooked process on Arctic shelves and highlights the role of the Arctic as a significant global nitrogen sink.

[1] The University of Texas at Austin, Marine Science Institute, 750 Channel View Drive, Port Aransas, Texas 78373, USA. † Present address: National Oceanographic and Atmospheric Administration, National Center for Coastal Ocean Science, 101 Pivers Island Road, Beaufort, North Carolina 28516, USA. Correspondence and requests for materials should be addressed to N.D.M. (email: nathan.mctigue@noaa.gov).

Since nitrogen is a key nutrient supporting marine primary production, the controls on its removal and recycling pathways are critical to understanding global biogeochemical cycles. Nitrogen loss processes that transform fixed bioavailable nitrogen to inert dinitrogen gas ($N_2$) are the prominent global nitrogen sink[1]. An estimated 44% of the world's fixed $N_2$ losses occurs on continental shelves[2], which constitute only 7.5% of the global seafloor[3]. The Arctic Ocean and its surrounding shallow seas represent a disproportionate 18% of the world's continental shelves, despite only covering 2.6% of the global ocean[3,4]. Therefore, Arctic shelves may have a critical role in the global nitrogen cycle, accounting for almost 10% of global $N_2$ losses.

The dominance of specific nitrogen transformation pathways in marine sediments determines whether a system removes fixed nitrogen or retains it within the ecosystem. Canonical heterotrophic denitrification (hereafter, denitrification), the microbially mediated anaerobic transformation of nitrate ($NO_3^-$) to inert $N_2$ gas, is the major removal pathway of nitrogen and is widespread in continental shelf sediments[5,6]. Anaerobic ammonium oxidation (anammox) is a chemoautotrophic pathway that removes inorganic nitrogen able to fuel primary production ($NH_4^+$ and $NO_2^-$) as $N_2$. Although anammox is not an ubiquitous process[7], it can rival denitrification by producing up to 80% of the total $N_2$ under certain conditions[8,9]. Dissimilatory nitrate reduction to ammonium (DNRA) is a microbially mediated anaerobic pathway that retains bioavailable nitrogen within the system, using $NO_3^-$ and producing $NH_4^+$. Lastly, nitrogen fixation represents the microbial conversion of inert $N_2$ to bioavailable nitrogen. The controls on the environmentally variable nitrogen fixation process are not well understood but appear to be regulated by nitrogen, phosphorus and iron availability[10]. Without nitrogen fixation, an ecosystem relies on either allochthonous delivery of dissolved inorganic nitrogen (DIN) or the recycling and release of bioavailable nitrogen from sediments by DNRA, ammonification (heterotrophic transformation of organic matter to $NH_4^+$), or nitrification (aerobic microbially mediated transformation of $NH_4^+$ to $NO_3^-$).

The extent and geochemical controls of denitrification and anammox are not necessarily predictable. Both denitrification and anammox require suboxic conditions but differences in the concentrations of $NO_3^-$, $NO_2^-$ and $NH_4^+$, the presence of $H_2S$, and the quantity and quality of organic carbon (OC) may favour one pathway over the other[1,11–15]. Thus, the fraction (%) of total $N_2$ production attributable to anammox ($ra$) is variable across environmental gradients[1,12]. Shallower locations that receive higher and more labile OC loads favoured denitrification, while more stable and organic-poor sites, situated deeper than 700 m, exhibited $ra$ of at least 67%, albeit with lower absolute rates of both potentially competing processes[14]. $NO_3^-$ concentrations often correlate positively with $ra$, although multiple environmental factors in concert, including sediment reactivity, can explain $ra$ better than any one sole factor[15]. Temperature may also have a regulatory role for the two processes since the optimum temperature for polar-adapted denitrifying bacteria is 20–25 °C, while the optimum temperature for polar-adapted anammox bacteria is lower, at 9–12 °C (refs 16,17). While these data suggest that anammox may have a competitive advantage in perennially cold Arctic sediments, results from temperature block experiments showed $ra$ at 2.5 °C was only ∼22% (ref. 16).

Abiotic environmental factors do not control biogeochemistry entirely. Biogeochemical processes can be enhanced by the activities of bioturbating and tube-dwelling sediment infauna[18]. Since burrows effectively increase the sediment-water interface,

and some infauna actively ventilate their burrows subsequently amplifying solute exchange, bioirrigation can enhance denitrification rates two to fivefold[18–20]. While abundance of tube dwelling organisms can suffice to describe bioturbation effects[20], a useful index of potential bioturbation (BPc) incorporates species abundance, biomass, mobility and sediment reworking mode into one metric[21].

The Chukchi Sea is considered the Pacific 'gateway' sea to the rest of the Arctic Ocean, serving as one of two connections to the global ocean. Through the Bering Strait, the Chukchi Sea receives northerly advected deep Pacific water containing relatively high concentrations of $NO_3^-$, which subsequently fuel some of the highest primary production in all of the Arctic[22,23]. A large fraction of this primary production is deposited onto sediments, ultimately supplying food for benthic macrofaunal and microbial food webs[24,25]. The Hanna Shoal region is an ecological hotspot since currents eddy around the shoal and deposit organic matter to the seafloor[26–28]. The northeast Chukchi Sea Shelf, which may contain >100 billion barrels of oil and >2.8 trillion $m^3$ of natural gas, is an area of possible future development[29]. Understanding the biogeochemical function of the area is urgent. Relatively high rates of denitrification were measured in the neighbouring Bering Sea shelf, which suggests that cold, high latitude sediments can have an important role in nitrogen cycling and ultimately modulate global elemental cycles[30].

The goal of this research was to quantify denitrification, anammox, DNRA and nitrogen fixation rates simultaneously in sediments near Hanna Shoal in the Chukchi Sea, and then relate the interstation variability of these rates to environmental factors that may facilitate and regulate the processes along the shelf. Previous measurements of the net $N_2$ flux from sediments indicate that Chukchi Sea sediments are a sink for bioavailable nitrogen[31–33]. However, without using $^{15}N$ isotope addition experiments, these studies could not simultaneously measure multiple pathways, distinguish denitrification from anammox, or distinguish denitrification fuelled by $NO_3^-$ from the water column versus nitrification within the sediments. Furthermore, quantifying the recycling pathway of DNRA elucidates the degree that sediments contribute DIN for primary production versus serve as a net sink of nitrogen. These four nitrogen transformation processes have seldom been measured simultaneously, and never in the Arctic[34–36]. To our knowledge only denitrification rates have been reported in the Chukchi Sea, and DNRA rates have not yet been reported anywhere in the Arctic at detectable levels. Here we show that denitrification is the major removal pathway in the Chukchi Sea, dominating anammox and DNRA, while nitrogen fixation was undetectable. The nitrogen cycling rates are highly correlated with BPc, whereas we found no relationship with abiotic factors. Extrapolating our results, we argue that Arctic shelves have a significant role in the global nitrogen cycle.

## Results

**Site characteristics.** Physio-chemical characteristics of sampling stations were relatively uniform across stations (Table 1). Sampling stations (Fig. 1) ranged between depths of 41 and 66 m. Bottom water temperatures (− 1.7 to − 1.6 °C) and salinities (32.7–32.8) were similar across all stations. Surface (0–2 cm) sediment C:N ratios ranged between 8.2 and 9.6. Sediment OC concentration spanned 0.65–1.79%, whereas total nitrogen concentration ranged between 0.09 and 0.23%. Porewater $NH_4^+$ (0–5 cm) ranged from the lowest concentrations at CBL13 ($38.7 \pm 2.1 \mu M$) to the highest concentrations at CBL11 ($85.5 \pm 35.2 \mu M$). Ambient $NO_3^-$ concentrations of bottom water ranged between 4.7 and 6.8 μM, while $NH_4^+$ concentrations were

**Table 1 | Summary of station characteristics.**

|  | CBL11 | CBL13 | H17 | H29 | H33 |
|---|---|---|---|---|---|
| Date occupied (month/date/year) | 8/2/2013 | 8/6/2013 | 8/4/2013 | 8/9/2013 | 8/10/2013 |
| Depth (m) | 47 | 50 | 41 | 66 | 50 |
| Latitude (°N) | 72.1033 | 71.2982 | 71.9913 | 71.9286 | 71.8228 |
| Longitude (°W) | 165.4556 | 161.6887 | 163.3834 | 158.3279 | 159.6097 |
| Sediment C:N (mol:mol) | 8.2 | 9.0 | 8.2 | 8.6 | 9.6 |
| Sediment OC (%) | 0.90 | 1.79 | 0.65 | 1.51 | 1.24 |
| Sediment TN (%) | 0.14 | 0.23 | 0.09 | 0.21 | 0.15 |
| Porewater $NH_4^+$ (μM) | 85.5 ± 35.2 | 38.7 ± 2.1 | 70.5 ± 6.8 | 61.6 ± 3.8 | 76.5 ± 7.7 |
| Bottom salinity | 32.7 | 32.7 | 32.7 | 32.8 | 32.7 |
| Bottom temperature (°C) | −1.6 | −1.7 | −1.6 | −1.6 | −1.7 |
| Bottom $NO_3^-$ (μM) | 5.2 | 4.7 | 5.1 | 5.5 | 6.8 |
| Bottom $NH_4^+$ (μM) | 2.6 | 1.6 | 1.7 | 1.4 | 2.4 |
| Bottom DO (% saturation) | 83.9 | 74.3 | 82.8 | 79.2 | 75.5 |

Sediment C:N, organic carbon (OC), and total nitrogen (TN) for top 2 cm. Sediment porewater $NH_4^+$ (mean ± s.e.m.) collected from top 5 cm ($n = 2$). Bottom water temperature, salinity, nutrients and dissolved oxygen (DO) measured ~3 m from seafloor.

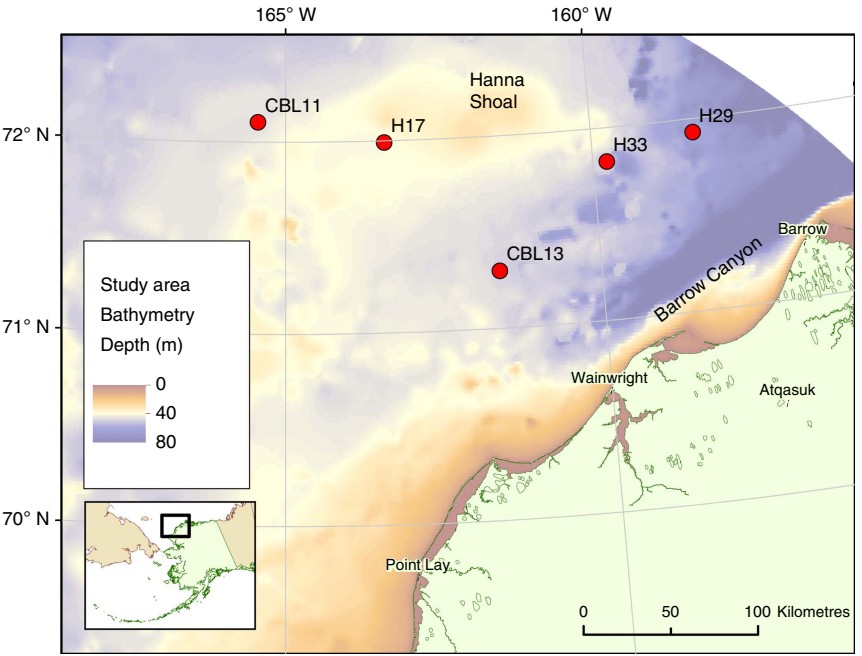

**Figure 1 | Location of sampling stations in the northeast Chukchi Sea.** Red circles represent the subset of stations from the Hanna Shoal Ecosystem Study where cores were collected for nitrogen biogeochemistry experiments. Colour gradient represents bathymetry. Landmass (light green) is northwestern Alaska, USA with villages labelled.

lower, between 1.4 and 2.6 μM. Dissolved oxygen content of bottom water reflected well oxygenated water, ranging from 74.3 to 83.9% saturation.

**Biogeochemical rates.** Using the isotope pairing technique allows us to report all nitrogen transformation processes in terms of the ambient $^{14}$N rates[37], which is critical for two reasons: first, they reflect *in situ* rates of the study area and are not influenced by the addition of $^{15}$N tracers, and second, the different rates are intercomparable. Denitrification, anammox and DNRA are reported, respectively, as $D_{14}$, $A_{14}$ and $DNRA_{14}$.

$D_{14}$ exhibited interstation variability across Hanna Shoal. The lowest rates (mean ± s.e.m.) were measured at H29 ($4.5 \pm 0.5$ μmol N m$^{-2}$ h$^{-1}$), while CBL13 had the highest rate of $20.4 \pm 3.0$ μmol N m$^{-2}$ h$^{-1}$ (Table 2, Fig. 2a). Across all five stations, the mean $D_{14}$ rate was $9.3 \pm 6.3$ μmol N m$^{-2}$ h$^{-1}$. $NO_3^-$ produced from sediment nitrification fuelled 58–92% of denitrification ($D_n$; Table 2, Fig. 2b). Although rates of

denitrification of the $^{15}NO_3^-$ tracer ($D_{15}$) are not reflective of *in situ* denitrification, they do show potential denitrification capabilities of the denitrifying microbial community[38]. $D_{15}$ and $D_{14}$ were correlated ($r^2 = 0.81$), thus $D_{15}$ mirrored $D_{14}$ rates in that the lowest rates ($1.5 \pm 0.5$ μmol N m$^{-2}$ h$^{-1}$) were measured at H29, whereas the highest rates ($20.4 \pm 3.5$ μmol N m$^{-2}$ h$^{-1}$) were observed at CBL13 (Table 2). Therefore, total potential denitrification ($D_{tot} = D_{14} + D_{15}$) peaked at $40.8$ μmol N m$^{-2}$ h$^{-1}$ in CBL13 sediments. $A_{14}$ rates were one to two orders of magnitude lower than $D_{14}$. The mean $A_{14}$ rate across stations was $0.22 \pm 0.02$ μmol N m$^{-2}$ h$^{-1}$ (Table 2, Fig. 2c). $A_{14}$ contributed between $1.4 \pm 0.1$ to $3.2 \pm 1.1\%$ of total $N_2$ production (Fig. 2d). The highest $ra$ values were observed where $D_{14}$ was low compared with other stations. $DNRA_{14}$, like $A_{14}$, was one to two orders of magnitude lower than $D_{14}$ (Table 2, Fig. 2e). The average $DNRA_{14}$ rate across all stations was $0.23 \pm 0.05$ μmol N m$^{-2}$ h$^{-1}$. $DNRA_{14}$ was not detectable

**Table 2 | Station summary of biogeochemical rates.**

| Station | n | $D_{15}$ | $D_{14}$ | $D_n$ (rate) | $D_n$ (%) | $A_{14}$ | ra | $DNRA_{14}$ | SOD |
|---|---|---|---|---|---|---|---|---|---|
| CBL11 | 6 | 7.8 ± 0.5 | 7.3 ± 0.4 | 5.5 ± 0.3 | 75 ± 0.4 | 0.24 ± 0.03 | 3.2 ± 0.4 | 0.09 ± 0.06 | −207 ± 7.2 |
| CBL13 | 4 | 20.4 ± 3.5 | 20.4 ± 3.0 | 16.1 ± 2.2 | 79 ± 0.7 | 0.29 ± 0.05 | 1.4 ± 0.1 | 0.53 ± 0.18 | −436 ± 49.5 |
| H17 | 6 | 6.9 ± 1.1 | 5.2 ± 0.7 | 3.6 ± 0.5 | 69 ± 3.1 | 0.15 ± 0.02 | 2.9 ± 0.4 | 0.25 ± 0.13 | −230 ± 17.1 |
| H29 | 4 | 1.5 ± 0.5 | 4.5 ± 0.5 | 4.1 ± 0.4 | 92 ± 2.3 | 0.10 ± 0.04 | 2.4 ± 1.6 | 0.11 ± 0.12 | −102 ± 26.8 |
| H33 | 4 | 18.6 ± 4.0 | 11.7 ± 1.8 | 6.7 ± 0.9 | 58 ± 3.6 | 0.33 ± 0.08 | 3.2 ± 1.1 | 0.18 ± 0.09 | −548 ± 28.4 |

Rates (mean ± s.e.m.) of denitrification of $^{15}NO_3^-$ ($D_{15}$), denitrification of $^{14}NO_3^-$ ($D_{14}$), the proportion of $D_{14}$ that is coupled to nitrification ($D_n$), anammox ($A_{14}$), proportion of $N_2$ produced by anammox (ra), dissimilatory nitrate reduction to ammonium (DNRA) and sediment oxygen demand (SOD). $D_{15}$, $D_{14}$, $D_n$ (rate), $A_{14}$ and $DNRA_{14}$ expressed as $\mu mol\,N\,m^{-2}\,h^{-1}$. SOD is expressed as $\mu mol\,O_2\,m^{-2}\,h^{-1}$. $D_n$(%) and ra are percentages.

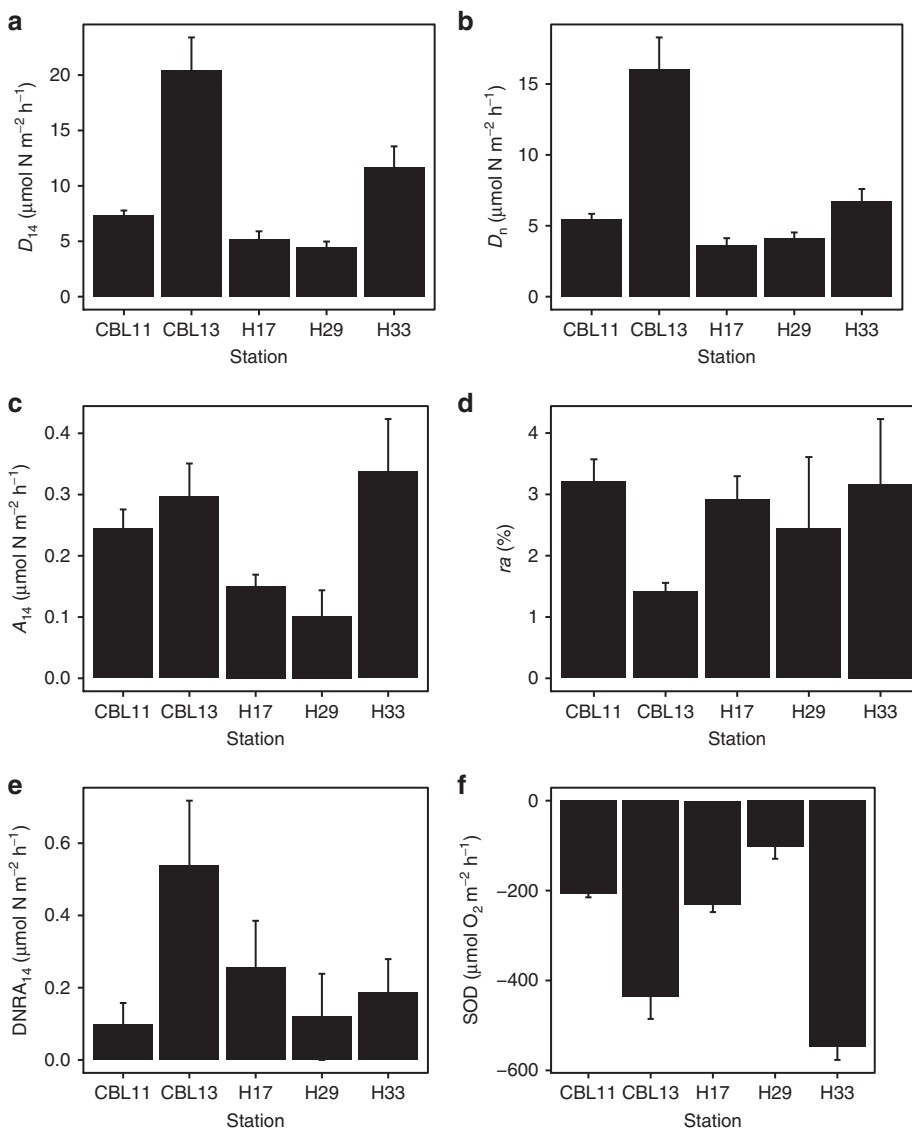

**Figure 2 | Biogeochemical rates from sampling stations.** Rates (mean ± s.e.m.) of (**a**) Denitrification of $^{14}NO_3^-$ ($D_{14}$), (**b**) amount of $D_{14}$ fuelled by nitrification-derived $NO_3^-$ ($D_n$), (**c**) anammox ($A_{14}$), (**d**) proportion of $N_2$ produced by $A_{14}$, (**e**) dissimilatory $^{14}NO_3^-$ reduction to $NH_4^+$ ($DNRA_{14}$) and (**f**) sediment oxygen demand (SOD). Note different y axis scales.

during some sampling events, but was also as high as $0.80\,\mu mol\,N\,m^{-2}\,h^{-1}$ at CBL13. The mean sediment oxygen demand (SOD) from dark incubations was $-290.8 ± 33.3\,\mu mol\,O_2\,m^{-2}\,h^{-1}$ (Table 2, Fig. 2f). This estimate varied across stations where the highest rates measured at H33 were more than five times greater than the lowest rates, which were observed at H29. Nitrogen fixation was not detected at any station.

**Relationship between rates and environmental variables.** The sampling design of this study conserved several environmental factors that can control nitrogen transformations. Temperature, salinity, depth and latitude were uniform across the sampled stations and could not drive heterogeneous interstation variability of $D_{14}$ (Table 1). Bottom water $NO_3^-$, $NH_4^+$ and dissolved oxygen concentrations, porewater $NH_4^+$ concentrations and sediment carbon lability indicators (OC content and C:N) were

**Table 3 | Pearson correlation matrix relating station characteristics to biogeochemical process rates.**

| | $D_{14}$ | $A_{14}$ | $DNRA_{14}$ | SOD | $D_n$ (%) |
|---|---|---|---|---|---|
| BPc | **0.97** | 0.59 | **0.95** | 0.66 | − 0.09 |
| Total abundance | **0.95** | 0.56 | **0.97** | 0.64 | − 0.11 |
| Polychaete abund. | **0.92** | 0.43 | **0.95** | 0.46 | 0.14 |
| Total biomass | 0.78 | **0.94** | 0.54 | **0.99** | − 0.72 |
| Species richness | **0.86** | 0.44 | **0.98** | 0.62 | − 0.14 |

Reported numbers are correlation coefficients (*r*-values). Bold values are correlations where $P < 0.05$. BPc, community-wide bioturbation potential; Total abundance, mean infaunal abundance; $D_{14}$, denitrification of $^{14}NO_3^-$; $A_{14}$, anaerobic $^{14}NH_4^+$ oxidation; $DNRA_{14}$, dissimilatory $^{14}NO_3^-$ reduction to $NH_4^+$; SOD, sediment oxygen demand; $D_n$, proportion of $D_{14}$ fuelled by sediment nitrification. The following parameters were tested, but were not significantly correlated with biogeochemical process rates: bivalve abundance, amphipod abundance, C:N ratio of sediment (0–2 cm), sediment organic carbon concentration, site depth, temperature, porewater ammonium concentration, and bottom water nitrate, ammonium and dissolved oxygen concentrations.

variable among stations but showed no significant trends with any biogeochemical process rates (Table 3).

## Discussion

Denitrification was the dominant nitrogen removal process, with rates exceeding anammox at all stations during summer sampling (Fig. 2a,c). Such low *ra* (1.4 ± 0.1 to 3.2 ± 1.1%) for the study area might be explained by examining the sources of OC to the sediment. During the spring and summer in the Chukchi Sea, a large fraction of the high primary production reaches the seafloor ungrazed as a seasonal pulse[22,39]. This labile carbon source fuels the benthic food web, including heterotrophic microbial processes like denitrification. Paradoxically, an unremarkable standing stock of OC (0.65–1.79%), though typical of Arctic shelves[40,41], was observed at the time of sampling. Standing stock may have poorly assessed the dynamic delivery and subsequent breakdown of organic matter on the seafloor given the relatively high rates of oxygen consumption (Fig. 2f). Regardless, the seasonally high organic matter deposition likely benefits denitrifiers over anammox bacteria in summer months since denitrification typically favours sites with higher OC delivery versus anammox, which tends to occur in organic-poor conditions[8,42,43]. Anammox may be more prevalent down-slope or in the Arctic Ocean basin at deeper depths where organic matter delivery is more attenuated[44] and where denitrification occurs at lower rates[32], following the trend of *ra* observed in other shelf-slope transects[14]; however, further studies are required to validate this hypothesis. Previous research suggests that net $N_2$ flux from sediments did not change significantly between ice-free and ice-covered seasons on the Alaskan Arctic continental shelf, indicating a lack of seasonality in the process[32,40]. How *ra* might vary over an annual cycle as OC quantity and quality change in the Arctic is unknown. Macrofauna subsist year-round from the intense and seasonally pulsed primary production since polar shelf sediments are long-term repositories of organic matter[25,45]. Perhaps the microbial food web maintains consistent year-round levels of activity from this 'food bank'.

DNRA, which is hypothesized to favour organic-rich sediments[42], may not be prevalent in Chukchi Sea shelf sediments that typically have <2% OC content[23]. Previous attempts by Gihring *et al.*[41] were unable to measure DNRA at detectable levels in Svalbard fjords in summer (depth = 51–211 m, temperature = 1–2 °C, 0.30–1.45% OC). We present here the first measurable rates of DNRA in the Arctic, albeit all were <1 µmol N m$^{-2}$ h$^{-1}$. Others have suggested that temperature appears to be important for DNRA, as temperature modulates sediment oxygen consumption that ultimately provides a favourable anoxic environment[34,46]. Although

DNRA has been reported as the primary $NO_3^-$ reduction pathway in tropical estuaries[7] and negligible in cold sediments[47], a linear temperature gradient cannot solely explain its global prevalence. For example, only one-third of studies in temperate marsh and tropical mangrove systems report DNRA rates that exceed denitrification, which suggests other factors besides temperature have a role in its variability[48]. A high ratio of OC versus $NO_3^-$ availability has been useful in predicting DNRA prevalence[42,43,48], but not without exceptions. DNRA dominated denitrification in hypoxic sediments in the Baltic Sea, where sediment organic matter was low compared with relatively high $NO_3^-$ concentrations[49]. Large salinity fluctuations can also favour DNRA over denitrification[38,50]. At our study site, low sediment OC accumulations, year-round $NO_3^-$ availability in bottom waters, and a lack of drastic salinity changes likely controlled DNRA rather than temperature since SOD was high. Still, denitrification was one to two orders of magnitude greater, and DNRA does not appear to be quantitatively important in the Hanna Shoal sediments.

No nitrogen fixation was detected in the Chukchi Sea sediments. Some fixation is photoautotrophic, which would have ceased in our dark incubations. Low rates of nitrogen fixation, measured by acetylene reduction, occurred in Arctic continental shelf sediments including 0.24 µmol N m$^{-2}$ h$^{-1}$ in the Canadian Beaufort Sea[51], 0.04–0.33 µmol N m$^{-2}$ h$^{-1}$ in Elson Lagoon in the Alaskan Beaufort Sea[52], and 0–0.83 µmol N m$^{-2}$ h$^{-1}$ in Svalbard fjords[41]. In comparison to $N_2$ production, these rates are inconsequential. For example, the highest nitrogen fixation rates reported by Gihring *et al.*[41] were only ∼6% of the combined $N_2$ production they measured from denitrification and anammox.

High water column primary production in the Chukchi Sea is driven by $NO_3^-$ upwelled from the deep Pacific Ocean onto the Bering Shelf and advected northward through the Bering Strait[22]. Yet the possibility of sediments recycling DIN for primary production remained plausible. Sediments contributed little $NH_4^+$ to the water column via DNRA (Fig. 2e), but the amount of $NH_4^+$ contribution from sediment remineralization was not tested and could be a source of DIN to the overlying water. Hanna Shoal sediments were a strong sink of nitrogen with denitrification dwarfing DNRA. This corroborates observations from the few other studies that have examined nitrogen cycling in the Chukchi Sea, which showed a net $N_2$ loss from sediments[31–33]. We refrain from directly comparing our denitrification data to rates from these previous studies that used different methodology. The isotope pairing technique consistently produces rates that are less than those measured by $N_2$:Ar flux, especially where bioturbation is prominent[53]; therefore, direct comparison would be misleading. Regardless of comparability, all measurements from multiple studies in the Chukchi Sea converge on the same overall trend that the sediments are strong sinks for fixed nitrogen.

A large fraction of $NO_3^-$ that is denitrified to inert $N_2$ is coupled to nitrification ($D_n$) of porewater $NH_4^+$ (Table 2). Given moderately high porewater $NH_4^+$ concentrations and low bottom water $NO_3^-$ concentrations (Table 1), we demonstrate that at least half of the DIN that fuels denitrification is from organic matter remineralization and subsequent nitrification in the sediments, and not from the water column $NO_3^-$ that could potentially fuel primary production. Although we did not measure it directly, sediment nitrification must be relatively active given that $D_n$, which hinges on nitrification to provide $NO_3^-$, was 58–92%.

While our sampling stations spanned the northeast Chukchi Sea shelf near Hanna Shoal, many environmental parameters that have been reported to affect nitrogen cycling (for example, temperature, salinity, depth, season) were uniform and could not

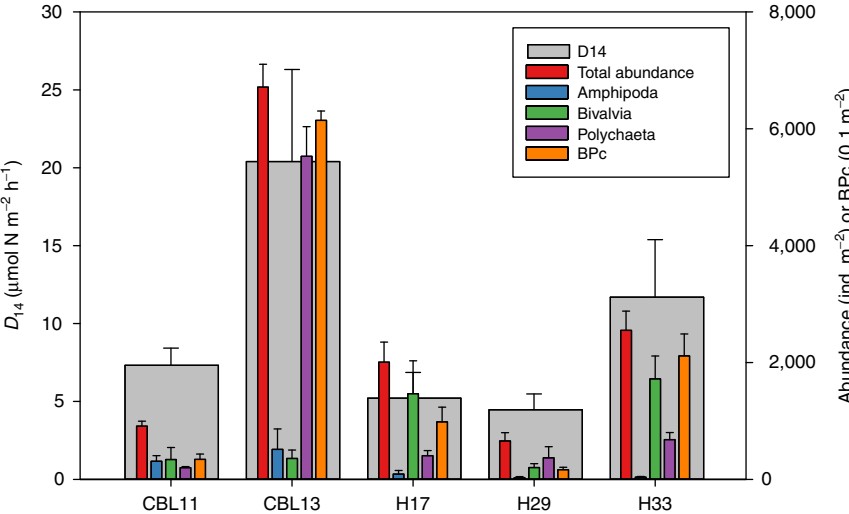

**Figure 3 | Station denitrification rates overlain by infauna abundance and bioturbation potential.** Mean (±s.e.m.) abundance and index of potential bioturbation (BPc) compared with denitrification ($D_{14}$) rate at each station. Total infaunal abundance and BPc were calculated from all taxa collected at each station.

control the variation in biogeochemical rates (Table 1). Temperature in the incubation chamber was maintained at 4 °C, which was slightly warmer than the *in situ* seafloor temperature. While this could increase the biogeochemical rates, though not likely the partitioning between pathways, the interstation rate differences were not temperature-induced. Moreover, some parameters that were variable among stations (for example, C:N, bottom water $NO_3^-$, porewater $NH_4^+$) showed no correlation with nitrogen transformation and oxygen demand rates (Table 3).

Organic matter stoichiometry (that is, elemental C:N ratios) is a strong predictor of *ra* in oxygen-deficient zones (ODZs) in the open ocean where anammox is likely $NH_4^+$ limited[13]. Since labile organic matter (exhibiting lower C:N ratios) will yield more $NH_4^+$ than more refractory organic matter (exhibiting higher C:N ratios) upon heterotrophic breakdown, Babbin *et al.*[13] demonstrated that *ra* was directly related to organic matter stoichiometry. It is difficult to directly compare the results of the current study, which presents coarse measures of sediment C:N ratios (0–2 cm, Table 1) amalgamating the organic-rich surface sediments with the narrow horizon of the nitrate reduction zone, to the trends reported by Babbin *et al.*[13]. Sediments may also violate some of the assumptions made in using organic matter stoichiometry to predict *ra*. Unlike ODZs, in sediments there is often incomplete oxidation of organic matter, $NH_4^+$ can accumulate in porewater (Table 1), and $NH_4^+$ produced in the nitrate reduction zone can be advected into the oxic nitrification zones or efflux into the overlying water. While our experimental design did not explicitly test if the same controls on *ra* in ODZs apply to sediments, they are likely more related to alternative environmental controls such as temperature[15], $NO_2^-$ and $NO_3^-$ availability[15,16], or sulfide concentration[11,12] given appreciable concentrations of porewater $NH_4^+$ in sediments. Lastly, the presence of benthic infauna could further alter the stoichiometry of sediment organic matter in unpredictable ways by altering the quality and quantity of OC and either concentrating or removing DIN from the nitrate reduction zone.

Because abiotic factors did not correlate with nitrogen transformation processes, biotic factors instead were explored as regulators, specifically infaunal bioturbation since it has previously been shown to enhance P (ref. 54) and N (refs 20,55,56) cycling in other benthic ecosystems. Infauna were collected and catalogued for a parallel research component of the Hanna Shoal Ecosystem Study following the methodology described by

Schonberg *et al.*[27]. Infaunal abundance and biomass data publicly available at http://arcticstudies.org/hannashoal/index.html (summarized in Supplementary Table 1) were explored as factors that might correlate with nitrogen transformation rates, and then were used to calculate a community index of potential bioturbation (BPc) for each station[21]. The BPc index weighs the abundance and biomass of bioturbating organisms higher than those that do not modify the sediment. Infauna abundance and biomass correlated highly with $D_{14}$, $A_{14}$, $DNRA_{14}$ and SOD rates within our study site (Table 3, Fig. 3). BPc showed the strongest relationship with $D_{14}$ and $DNRA_{14}$ ($r = 0.97$ and 0.95, respectively) of all parameters measured. Polychaete abundance correlated with $D_{14}$ and $DNRA_{14}$ ($r = 0.92$ and 0.95, respectively), although total abundance had a stronger relationship with the two processes ($r = 0.95$ and 0.97, respectively). $A_{14}$ and SOD correlated with infaunal biomass ($r = 0.94$ and 0.99), but not abundance. SOD co-varied with $A_{14}$ ($P < 0.05$, $r = -0.91$), suggesting anammox was most active in the least oxic sediments, possibly modulated by fauna respiration. Bivalve and amphipod abundance alone did not significantly relate to any processes.

The strikingly high infauna abundance at CBL13 was driven by the tube-dwelling polychaete *Maldane sarsi*, with densities of $5,057 \pm 484$ individuals $m^{-2}$, whereas the only other station containing *M. sarsi* was H17 at a density of $3.3 \pm 5.8$ individuals $m^{-2}$ (Supplementary Table 1). The dense population of the tubicolous polychaete *M. sarsi* at CBL13 physically amended the habitat for denitrifying and DNRA microbes (Supplementary Fig. 1). The tubes themselves functionally increased the sediment-water interface surface area, thus, creating more volume of sediment where denitrification or DNRA could occur. The tubes also acted as ventilated conduits for solute exchange, which would accelerate the delivery of either $NO_3^-$ from the water column to fuel DNRA and direct denitrification ($D_w$), or conversely, oxygen to fuel nitrification for $D_n$ (refs 19,20). Since $D_n$ was relatively high (58–92%), we postulate that infauna increased oxygen transportation and subsequently facilitated nitrification. The link between nitrification and fauna has been observed in the southeastern Chukchi Sea where relatively high rates of sediment nitrification (41.7–50.0 μmol N m$^{-2}$ h$^{-1}$) occurred in bivalve-dominated sediments relative to fauna-depauperate sediments[57].

Top-down feeding polychaetes (for example, *M. sarsi*) consume oxygen in the sediment, expanding conducive conditions for the suboxic processes of denitrification and

DNRA. Polychaete organic-rich secretions (for example, mucus, feces) are typically labile and converted readily to $NH_4^+$ to bolster nitrification[58]. At stations H17 and H33, bivalves represented ~70% of the total infauna abundance (Fig. 3, Supplementary Table 1). In this case, activities associated with bivalve deposit feeding, not the physical presence of tubes, enhanced denitrification. Therefore, a specific taxonomic group may not affect rates as much as the abundance and biomass of seafloor-modifying organisms. A recent study has demonstrated that bioturbation from meiofauna (invertebrates < 1 mm) doubled denitrification rates, and the additive effects of both meiofauna and macrofauna enhanced DNRA and methane efflux[56]. Moreover, macrofaunal injection of organic-rich biodeposits (feces and pseudo-feces) enhance heterotrophic denitrification by providing additional carbon substrate for heterotrophic processes[59].

While the infaunal bioturbation index (BPc) was the strongest correlating factor in our study, it is possible that the trend is not ubiquitous throughout the entire Chukchi Sea. For example,

Braeckman et al.[55] recently showed that infaunal abundance was a crucial predictor of biogeochemical processes at some sites in the North Sea, while abiotic factors were more important in muddy sediments.

To compare our measured rates of $D_{14}$, $A_{14}$ and $DNRA_{14}$ to other systems globally, mean rates (with standard deviations if given) were plotted against the temperature at which they were collected, or incubated if provided (Fig. 4a, Supplementary Table 2). In order to make meaningful comparisons, our literature review only included marine studies where sediment was not experimentally amended except for the addition of $^{15}N$ tracers, and denitrification rates were expressed as $D_{14}$ (refs 37,60). If studies reported rates on a volume or mass basis, they could not be directly compared with our areal rates, but their proportions of $NO_3^-$ reduction were deemed valid for comparison (Fig. 4b). The $D_{14}$ rates from the Chukchi Sea sediments fell within the range of the literature surveyed (Fig. 4a). Rates from our high latitude study and those from work conducted in Svalbard (~80 °N, 0–2 °C) demonstrated that perennially cold Arctic shelves are at least as important for denitrification as continental shelves in lower latitudes. $A_{14}$ and $DNRA_{14}$ rates were all relatively low in the literature surveyed but of similar magnitude to the Chukchi Sea rates, and no discernible trend with temperature was apparent. There was an indication that ra can have an inverse relationship with temperature (Fig. 4b) as demonstrated by Brin et al.[15], but there are many instances where ra was low at low temperatures. Comparatively, the Hanna Shoal region was consistently dominated by denitrification, with low ra and relative DNRA proportions. Temperature block experiments have demonstrated that denitrification rates of Arctic-adapted microbes are highest at temperatures between 20 and 25 °C (ref. 17), but many other factors in situ can regulate the process. Thus, in the literature surveyed there is no clear temperature regulation of denitrification, anammox, or DNRA. This evidence suggests that temperature is not a limiting factor for nitrogen cycling, and the disproportionately large continental shelf area surrounding the Arctic Ocean is important for global modulation of elemental cycling.

Biogeochemical rate measurements are logistically complex and time-consuming, and often a few stations must represent an expansive area. However, exploring correlative parameters can help reduce the error associated with interpolating data within a study area by providing a finer resolution to estimate the process. For example, in their study that transected shelf, slope and basin, Chang and Devol[32] derived a relationship between net $N_2$ flux, depth and export production for the Chukchi Sea, and extrapolated the relationship to each Arctic sea to estimate a global contribution of 4–13% for Arctic denitrification. The authors emphasize the potential role of the Arctic Ocean in global denitrification and the need for much greater spatial and temporal coverage of nitrogen cycling measurements.

Similarly, this current study faced a common challenge in scaling up measurements to the entire region. We utilized the strong significant relationship between $D_{14}$ rates and BPc ($P < 0.01$, $r = 0.97$; Table 3) measured at Hanna Shoal to extrapolate our rates to the region. Calculating BPc across the northeast Chukchi Sea was accomplished by utilizing data from past multi-disciplinary, multi-year projects in the northeast Chukchi Sea, where >100 benthic stations were occupied, many of which include infauna abundance and biomass (http://www.arcticstudies.org). Using this previously reported data, we calculated BPc at over 50 stations and used the linear regression derived from our stations ($D_{14} = 0.0026 \times BPc + 4.76$) to predict $D_{14}$ rates across the Chukchi Shelf as a spatially heterogeneous surface (Fig. 5). This approach alleviates the need to coarsely estimate an area's denitrification contribution by using

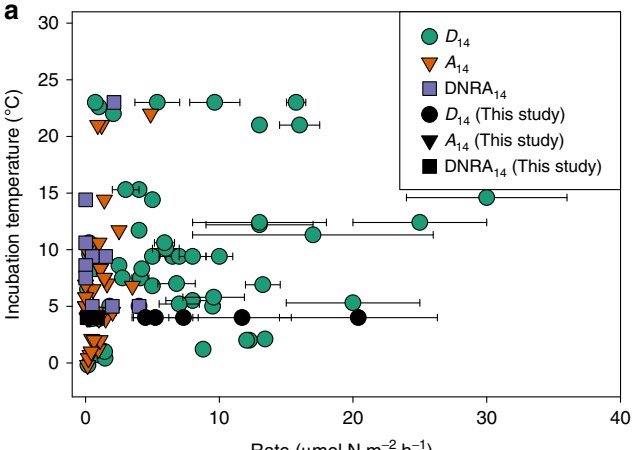

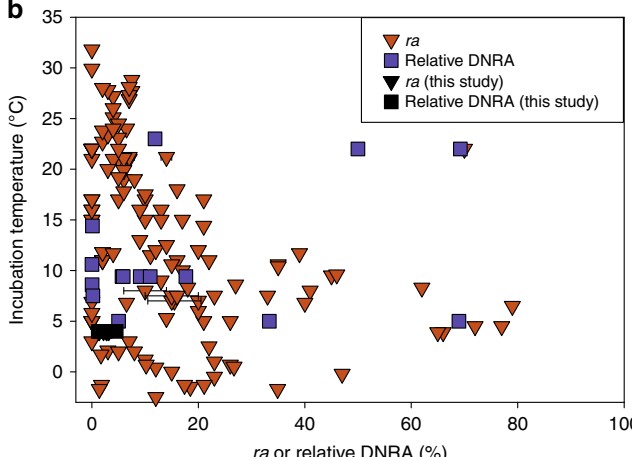

**Figure 4 | Survey of nitrogen transformation rates and ratios from literature and this study plotted by temperature.** (**a**) Rates of denitrification ($D_{14}$, circles), anammox ($A_{14}$, triangles) and $DNRA_{14}$ (squares), and (**b**) the proportion of anammox compared with total $N_2$ flux (ra) (triangles) or proportion of $NO_3^-$ that underwent $DNRA_{14}$ compared with denitrification (squares) plotted by temperature. Rates from this study (black symbols) were compared with surveyed literature. References and values are listed in Supplementary Table 2. All values were determined using the isotope pairing technique[37,60]. Rates and proportions are mean values and error bars are s.d., if provided.

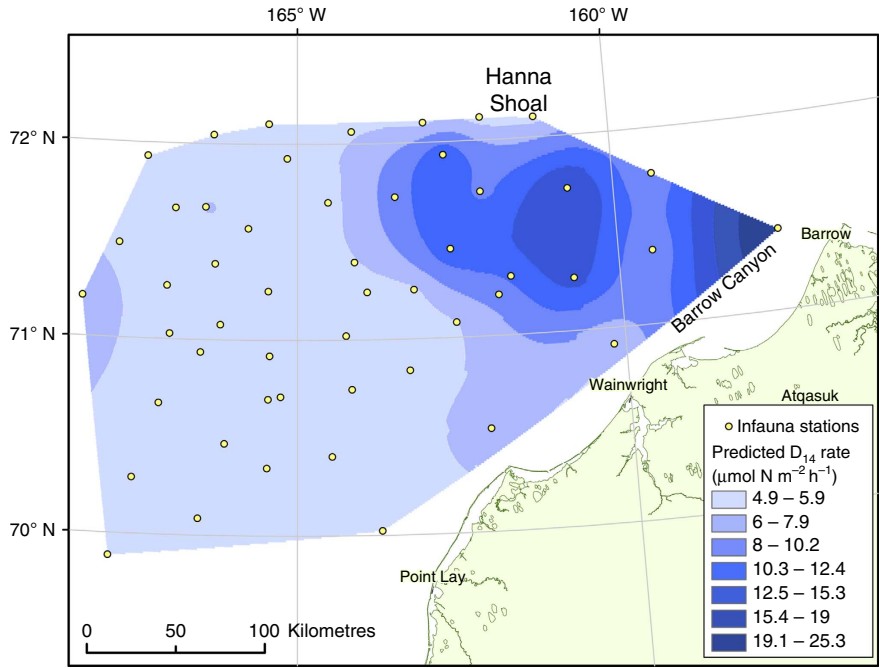

**Figure 5 | Predicted denitrification rates in the northeast Chukchi Sea.** Interpolated surface using empirical Bayesian kriging for predicted denitrification rate ($D_{14}$) across the northeast Chukchi Sea using the relationship $D_{14} = 0.0026 \times BPc + 4.76$. Infaunal abundance and biomass data were collected at all plotted stations for the calculation of a community-wide index of potential bioturbation (BPc).

| Technique | Correlation variable with $D_{14}$ | Spatial analysis | Modelling approach | Mean annual $D_{14}$ rate (Gmol N yr$^{-1}$ study site area$^{-1}$) |
|---|---|---|---|---|
| | | | **Table 4 | Three techniques used to estimate the annual denitrification rate from the study site.** | |
| 1 | None; used mean $D_{14}$ for all stations | None; used study area as a whole | Propagated error | $5.30 \pm 3.60$ |
| 2 | BPc for 51 stations | Empirical Bayesian kriging; 7 bins | Propagated error | $4.65 \pm 0.16$ |
| 3 | BPc for 51 stations | Empirical Bayesian kriging; 7 bins | Randomly generated rates in each bin for 100,000 iterations | $4.04 \pm 0.001$ |

a mean rate, which not only neglects the spatial variability of $D_{14}$ but may severely overestimate or underestimate the rates extrapolated across the study area.

Three different techniques were compared to estimate regional biogeochemistry and to demonstrate the advantage of correlating biogeochemical rates with other parameters (Table 4). First, we used the mean ($\pm$ s.d.) rate derived from our five occupied stations ($9.25 \pm 6.27 \, \mu mol \, N \, m^{-2} \, h^{-1}$) to extrapolate $D_{14}$ across the study area ($65,500 \, km^2$). For the second and third approaches, we used the relationship between $D_{14}$ and BPc to estimate denitrification at 51 additional stations in the northeastern Chukchi Sea. An interpolated, binned surface of denitrification rates (Fig. 5) was created by empirical Bayesian kriging. The second technique used the mean rate within each bin as a representative rate for the designated area. The sum of all bins was used as an integrated estimate of $D_{14}$ for the entire study area based on BPc. For the third technique, a randomly generated number within the specified range of each bin was assigned to each $m^2$ and summed across the area. This process was repeated for 100,000 iterations to obtain a mean rate.

The third technique produced a notably different estimate than Technique 1 and greatly reduced the uncertainty in the estimation (Table 4). If we assume that the strong relationship between $D_{14}$ and BPc occurs throughout the northeastern Chukchi Sea, then we would expect lower denitrification in much of the western part of the region since infauna abundance

and biomass, thus BPc, were lower there than in the eastern region of the study area (Fig. 5). The region southeast of Hanna Shoal and near the mouth of Barrow Canyon were predicted by our model to have the highest denitrification rates in the northeast Chukchi Sea due to the high benthic abundance of infauna, but spatially it is a small portion of the study area. Thus, in the first technique the higher rates measured at CBL13 represent one-fifth of the area and are equally weighted against the low rates measured at H29. In the third technique, the high rates ($> 20 \, \mu mol \, N \, m^{-2} \, h^{-1}$) only represented $\sim 5\%$ of the study area, whereas rates $< 5.3 \, \mu mol \, N \, m^{-2} \, h^{-1}$ are predicted to occur at $\sim 60\%$ of the seafloor. This exercise provides both a geographic distribution of potential denitrification activity and a better-approximated regional rate where little experimental data exist.

A major caveat for this procedure lies in the simplifying assumption that $D_{14}$ is equally related to BPc across the entire northeast Chukchi Sea, since it has been demonstrated that the strength of the relationship between BPc and biogeochemical rates may vary over an entire region[55]. Infaunal abundance integrates various important ecosystem drivers (for example, primary production, hydrography, sediment type) making it sufficiently robust to represent the complexities across an entire continental shelf. BPc may not be a globally transcendent predictor of denitrification, but was by far the most important factor in our study (Table 3), although all parameters should be

considered when creating relationships between rates and biotic or abiotic factors[15,55].

Using the lowest and highest rates of denitrification we modelled for the northeastern Chukchi Sea (Fig. 5) as bounds for rates possible in Arctic shelves, a range of annual denitrification was extrapolated for the entire Arctic shelf area ($5.052 \times 10^6 \, km^2$). Multi-seasonal work has indicated no difference between summer and winter/spring denitrification[32,40], so we assume the rates here are maintained over the course of a year. Annual rates for the Arctic shelves were constrained between 2.8 and 15.7 Tg N yr$^{-1}$, which account for between 0.5 and 2.7% of global denitrification (573 Tg N yr$^{-1}$; ref. 2). These annual rates account for 0.8 to 4.6% of marine denitrification despite only covering 1.4% of the global ocean[2,4]. Rates measured by the isotope pairing technique are often lower than those quantified by net $N_2$ flux methods[61], so these estimates are conservative compared with approaches that use the latter methodology. The higher range predicted here overlaps with other trans-Arctic annual denitrification estimates (6–29 Tg N yr$^{-1}$; ref. 32). Although these estimates are relatively coarse due to the paucity of actual sampling that has occurred in the Arctic, the results indicate that the Chukchi Sea is an active site of nitrogen cycling, and we suspect that the rest of the Arctic shelves might exhibit a similar range of rates. Since there was no discernible difference between denitrification rates in the Arctic compared with lower latitudes (Fig. 4a) and since Arctic shelves constitute almost 20% of global continental shelves, the perennially cold shallow shelves surrounding the Arctic Ocean should be recognized for their potentially critical role in the global nitrogen cycle.

## Methods

### Study site and sample collection.
These experiments were a component of the Hanna Shoal Ecosystem Study in the northeastern Chukchi Sea, Alaska (www.arcticstudies.org/hannashoal). Samples were collected between 2–10 August 2013 aboard the USCGC Healy at five stations that spanned Hanna Shoal from 158.3° W to 165.5° W but were latitudinally constrained between 71.3° N and 72.1° N (Fig. 1). Six intact cores (7.6 cm diameter; 20 cm deep) were collected from the seafloor of each station using a stainless steel HYPOX corer[34] that preserved the integrity of the sediment-water interface. After collection, cores were stored in a 4 °C environmentally controlled chamber until the experiment began (<4 h). All samples were subjected to the same treatments. Sixty litres of near-bottom water (within 3 m) was collected from each station by conductivity-temperature-depth (CTD) rosette cast and aerated continuously.

### Sediment core incubations.
Cores were incubated shipboard in a 4 °C environmentally controlled chamber to measure dissolved gas fluxes ($^{28}N_2$, $^{29}N_2$, $^{30}N_2$, $^{32}O_2$, $^{40}Ar$) and $^{15}NH_4^+$ production at the sediment-water interface. Incubations were conducted in a flow-through system in which bottom water was pumped over cores using a multi-channel peristaltic pump that maintained 1.0–1.4 ml min$^{-1}$ flow following Gardner and McCarthy[34]. An acetol plunger with a Viton o-ring was inserted into each core to ~5 cm from the sediment-water interface, which left ~230 ml of overlying water. Polyetheretherketone (PEEK) tubing connected each plunger's inlet port to unfiltered bottom water via a multi-channel peristaltic pump and the outlet port to a collection vessel. The 60 l of bottom water was partitioned into three separate carboys so that one carboy fed two replicate cores. One carboy was enriched with $^{15}N$-$NO_3^-$ (98.2% $^{15}N$-KNO$_3$) to final concentrations of 29.1–34.4 μM $NO_3^-$ depending on background concentrations (Table 1). The second carboy was spiked with $^{15}N$-$NH_4^+$ (99.9% $^{15}N$-NH$_4$Cl) to final concentrations between 15.1–17.7 μM $NH_4^+$. The third carboy was left unamended as a control treatment. Cores were wrapped in aluminum foil to ensure darkness throughout the experiment.

Inflow and outflow samples were collected once every 24 h for 4 days after an initial overnight pre-incubation to allow the experimental units to reach equilibrium. Samples for dissolved gases were collected in 13 ml Exetainers (Labco Limited, United Kingdom) by allowing vials to overflow from the bottom three times, ensuring no air bubbles were captured. Each collected sample was injected with 200 μl of saturated ZnCl$_2$ to halt microbial activity. Exetainers were capped, sealed with parafilm, and stored underwater at 4 °C. Approximately 30 ml of outflow water from the control and $^{15}NO_3^-$ treatment was collected to measure $^{15}NH_4^+$ production. Sample water was filtered immediately through a 0.2 μm filter into a Whirl-pak bag and then frozen at −20 °C. All samples were transported to the University of Texas Marine Science Institute for analyses.

### Benthic fluxes and nitrogen transformations.
Duplicate samples for dissolved gas concentrations ($^{28}N_2$, $^{29}N_2$, $^{30}N_2$, $^{32}O_2$ and $^{40}Ar$) were measured using membrane inlet mass spectrometry[62,63]. Analytical replicates ($n = 3$) for dissolved gases had a coefficient of variation <0.04%. $H_2O$ and $CO_2$ were removed cryogenically in line with a liquid nitrogen trap before dissolved gas introduction to the quadrupole mass spectrometer. The proportion of $^{15}NH_4^+$ from the total ammonium pool was determined by ammonium isotope retention time shift high-performance liquid chromatography[64]. Duplicate samples were injected three times to measure average retention time shift in samples incubated with $^{15}NO_3^-$ (coefficient of variation <5%). Dissolved gas and nutrient fluxes were calculated as

$$\text{benthic flux} \, (\mu mol \, m^{-2} \, h^{-1}) = (C_o - C_i) \times F / A \qquad (1)$$

where $C_o$ is the outflow concentration (μM), $C_i$ is the inflow concentration (μM), $F$ is the flow rate (l h$^{-1}$), and $A$ is core surface area (m$^2$). In this way, a positive flux is out of the sediment into the water column, which we refer to interchangeably as production. Denitrification rates were calculated based on the relationship between $^{14}N$-$N_2$ and $^{15}N$-$N_2$ production in the $^{15}NO_3^-$ treatments, following the isotope pairing technique[37]. Adding $^{15}NO_3^-$ increases overall denitrification rates, but the isotope pairing technique differentiates between denitrification of the added tracer and the in situ $^{14}NO_3^-$. One of the major assumptions of the isotope pairing technique is that the added nitrate did not induce higher rates of $^{14}NO_3^-$ denitrification by deepening the nitrate penetration depth in sediments. Testing this assumption would require measuring rates at multiple nitrate concentrations. While this was logistically impossible for our project, the assumption is met in most studies that use the isotope pairing technique at multiple nitrate concentrations.

Denitrification of $^{15}NO_3^-$ ($D_{15}$) and $^{14}NO_3^-$ ($D_{14}$) was calculated using the production of $^{29}N_2$ (p29$_{NO3}$) and $^{30}N_2$ (p30) from the $^{15}NO_3^-$ treatment using the equations

$$D_{15} = p29_{NO_3^-} + 2(p30) \qquad (2)$$

and

$$D_{14} = D_{15} \times [p29_{NO_3^-} / (2 \times p30)]. \qquad (3)$$

p29$_{NO_3^-}$ was determined after removing the minor $^{29}N_2$ contribution from anammox (modified from Risgaard-Petersen et al.[60]; see Supplementary Methods). The proportion of in situ denitrification from the overlying water column nitrate ($D_w$) was determined with the following equation from Nielsen[37]:

$$D_w = D_{15} \times [^{14}NO_3^-] / [^{15}NO_3^-] \qquad (4)$$

The concentration of $^{14}NO_3^-$ was determined from ambient bottom water, and $^{15}NO_3^-$ was determined as the difference between $NO_3^-$ concentration after addition of $^{15}N$ tracer and ambient levels. The remaining proportion of denitrification was, therefore, attributed to coupled nitrification-denitrification ($D_n$) and calculated as

$$D_n = D_{14} - D_w \qquad (5)$$

Total anammox ($A_{tot}$) estimates were calculated from production of $^{29}N_2$ in the $^{15}NH_4^+$ treatment (p29$_{NH_4^+}$) by

$$A_{tot} = p29_{NH_4^+} / F_A \qquad (6)$$

where $F_A$ is the ratio of $^{15}NH_4^+$ to total $NH_4^+$ in the overlying water. We report anammox as $A_{14}$, or the contribution of $^{14}NH_4^+$ and $^{14}NO_2^-$ to anammox since the $^{15}NH_4^+$ tracer may induce higher than natural rates, using the equation

$$A_{14} = A_{tot} - A_{15} \qquad (7)$$

where $A_{15}$ equals p29$_{NH_4^+}$. This approach assumes that labelled ammonium is not nitrified and subsequently denitrified (see Supplementary Methods). In the scenario of nitrification–denitrification of $^{15}NH_4^+$, anammox rates would be overestimated since anammox and denitrification would produce $^{29}N_2$.

Nitrogen fixation was measured using the procedure of An et al.[62], which uses the concentrations and fluxes of $^{28}N_2$, $^{29}N_2$ and $^{30}N_2$ in the $^{15}NO_3^-$ treatment to detect simultaneous fixation in the presence of denitrification.

DNRA was calculated by measuring the production of $^{15}NH_4^+$ (p$^{15}NH_4^+$) from the $^{15}NO_3^-$ treatment. We report DNRA as:

$$DNRA_{14} = (p^{15}NH_4^+ / F_N) - p^{15}NH_4^+ \qquad (8)$$

where $F_N$ is the proportion of $^{15}NO_3^-$ in the total nitrate pool[65].

$D_{14}$, $A_{14}$ and DNRA$_{14}$ are all conservative estimates of the true in situ rates since porewater, which could retain transformed nitrogen species, was not extracted from sediments. $D_{14}$ also can underestimate denitrification in the presence of bioturbators[53], but we allowed for an overnight equilibration for $^{15}NO_3^-$ and $^{15}NH_4^+$ incorporation into the porewater and measured rates over 4 days. Furthermore, DNRA$_{14}$ is conservative because cation exchange in sediments could retain $^{15}NH_4^+$, and we did not extract porewater from the flow-through experimental design. By using terms that are standardized to the ambient $^{14}N$ concentrations instead of those induced by the addition of the $^{15}N$ tracer, the rates can be compared as co-occurring processes.

**Sediment chemistry.** Samples for surface sediment characterization were collected separately from each station by using a van Veen grab ($0.1 \, m^2$). Samples for elemental C and N analysis were collected from the top 2 cm of sediment with a 10 ml syringe barrel and immediately transferred to a container and frozen at $-20 \, ^\circ C$. Two adjacent aliquots were sampled from the top 5 cm of undisturbed sediment by a 60 ml syringe barrel, placed in a Whirl-pak bag, and frozen at $-20 \, ^\circ C$ for sediment porewater $NH_4^+$ analysis.

To remove carbonates that would skew OC analysis, sediments were soaked in 1N HCl until bubbling stopped, then rinsed in deionized water and dried at $60 \, ^\circ C$ to a constant weight. Sediment was analysed on an elemental analyser (CE Instruments, NC 2500). C:N values are presented as mol:mol ratios.

**Nutrients.** Samples for $NO_3^-$ and $NH_4^+$ concentrations from station bottom water were prepared using scaled down volumes of sample and reactants suitable for a 96-well plate and read on a microplate spectrophotometer at 543 and 640 nm, respectively, following Mooney and McClelland[66]. $NO_2^-$ concentrations were determined by colorimetric spectrophotometric analysis[67]. $NO_x$ was measured after Cd reduction of $NO_3^-$ to $NO_2^-$ following a modified procedure of Jones[68]. $NO_3^-$ is reported as the difference of $NO_x$ and $NO_2^-$. $NH_4^+$ concentrations were measured colorimetrically with the indophenol blue method[69]. To obtain samples for porewater $NH_4^+$, sediment was thawed and centrifuged while still cold at 5,000 r.p.m. for 20 min. Supernatant was decanted and analysed on a Shimadzu UV-2401PC spectrophotometer.

**Statistical analyses and data treatment.** All statistics were computed using R 3.1.1 (www.r-project.org). To ensure the isotope tracers had reached the zone of $NO_3^-$ reduction after the overnight pre-incubation period, steady-state equilibrium was tested with a repeated-measures analysis of variance (*lme* function from the *nlme* package). For each station, dissolved gas or nutrient fluxes were the dependent variable, sampling time point was the group, and core was a random factor. The results underwent *post-hoc* tests using the *glht* function from the *multcomp* package that performed multiple comparisons for linear mixed effects models using Tukey's all-pair comparisons. Significantly different rates were excluded from analysis since they violated steady-state equilibrium assumptions. Before determining the rates of $A_{14}$ and $DNRA_{14}$, we determined if the respective production of $^{29}N_2$ and $^{15}NH_4^+$ was significantly higher than the control treatment using a one-way analysis of variance (*aov* function from the *MASS* package). A Pearson correlation matrix was created using mean rates and environmental parameters from each station by the *rcorr* function in the *Hmisc* package. Linear regressions were computed using the *lm* function from the *MASS* package. Values are reported as mean ± s.e.m. For all tests, alpha was set at 0.05. Spatial analysis was completed in ArcMap 10.3 (Esri, Redlands, CA) using the Geostatistical Analyst Toolbox function empirical Bayesian kriging, which creates an interpolated surface by accounting for the error in estimating the underlying semivariogram though iterative simulations[70].

**Data availability.** The data that support the findings of this study are available at http://arcticstudies.org/hannashoal or can be requested from N.D.M.

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

## Acknowledgements

We are grateful to captain and crew of the USCGC *Healy* for the safe and successful voyage during HLY1301. We thank Mark McCarthy and Anne Giblin for much help and direction in analysing samples and nitrogen cycling rate calculations. Susan Schonberg was the lead taxonomist for this project and provided infauna taxonomy and biomass data. We thank Kaijun Liu for analysing ammonium isotope retention time shift samples, and Kim Jackson, Nick Reyna and Audrey Wohlrab for assistance with nutrient analyses. Bill Adams and staff constructed the stainless steel HYPOX corer. Three anonymous reviewers greatly improved the clarity and focus of this manuscript. This component of the Hanna Shoal Ecosystem Study was funded by the U.S. Department of the Interior, Bureau of Ocean Energy Management (BOEM), Alaska Outer Continental Shelf Region, Anchorage, Alaska under Cooperative Agreement M11AC00007 with The University of Texas at Austin as part of the Chukchi Sea Offshore Monitoring in Drilling Area (COMIDA) Project and the BOEM Alaska Environmental Studies Program. This is UTMSI contribution 1713.

## Author contributions

N.D.M., A.K.H., and K.H.D. performed the field sampling. N.D.M., W.S.G., and A.K.H. carried out the experiments, laboratory analysis and data processing. The research was designed by all authors. N.D.M. wrote the paper but received essential guidance and feedback from all co-authors.

## Additional information

**Competing financial interests:** The authors declare no competing financial interests.

