## [Peer review file · Nature Communications]

Reviewers' comments:

Reviewer #1 (Remarks to the Author):

This paper documents the importance of different nitrate reduction process and N₂ fixation in shallow Arctic shelf sediments. The work is important for several reasons. First, the Arctic shelf is a large area making up nearly 20% of the continental shelf area and therefore the Arctic shelf may be very important in the overall oceanic N budget. Second is that some, but not all, previous studies have suggested that anammox may be more important in cold sediments than at lower latitudes. Additionally, this study ties in nitrogen cycling to bioturbation, something that is thought to be important but rarely examined in a quantitative way. Finally, this study has examined DNRA, something that has not been studied in the Arctic shelf previously.

The paper is well written but could be improved somewhat for clarity and there are a number of issues the authors need to comment on concerning methods.

What temperature were the experiments actually run at? The authors say in a 4°C environmental chamber. Were they at 4? The authors should make a note in the discussion that if so this was at least 5°C higher than bottom water temperature. I doubt this effected the partitioning but it possibly affected the overall rates? This needs to be mentioned and the possible impact of this on the overall extrapolation of rates considered.

The authors need to carefully comment on the method they used to measure anammox. There are now several methods in use to measure anammox. All have their advantages and disadvantages and there is considerable debate about the artefacts from each. In this case the authors added ¹⁵N labeled ammonium to the overlying water in a flow through system with oxic overlying water. The ¹⁵N ammonium could be directly used for anammox but it could also be nitrified and subsequently denitrified. This would cause an overestimation of the rates. Anammox could be underestimated if the labeled ammonium did not have time to reach the zone of anammox. Ammonium adsorbs to sediments so it diffuses more slowly than nitrate but this is more of a problem in short term incubation than the flow through study here. Finally, if a very significant amount of the added ammonium was nitrified and not denitrified, anammox could be greatly underestimated because no remaining ammonium would be available. This could be checked by looking at the ammonium and nitrate concentrations coming from the cores. These are not reported but measured? Assuming the label was not largely nitrified I suspect that the reported rates are maximum rates. This should be discussed.

Line 129 - D15 is "confounded" I agree with the authors' points but don't think it is quite the correct word. I think a better way to say it is that because of added NO₃, D15 don't reflect in-situ rates but have been used to assess denitrification potential. Did the D15 follow SOD better than D14 or track D14 closely? Given the low bottom water NO₃ I would have expected some stimulation of direct denitrification in sites with sufficient available C.

I liked the authors extrapolation to the entire shelf. They explain all of the caveats to the three methods. While there is a lot of uncertainty in the estimates it gives some order of magnitude estimate and will help future investigators to choose stations which will best be able to reduce the uncertainty.

I appreciate the effort that went into Figure 4 and supplementary table 2 but I wonder how effective it is, and it is graphically very unappealing. I think most of the points could simply be made in the text. As useful as Table S2 it does not contain other information (like SOD) that might be used to try to tease out relationships and it includes a variety of techniques (actual, potential) methods to measure some of these rates. I feel like this table and graph could be omitted here and used for a more comprehensive review paper or meta-analysis more effectively.

It would help to casual reader who firsts looks at the graphs to define BPC in the table and figure legends.

The correlation matrix (table 3) includes variable which did not really show any differences between stations (such as temperature). Surprisingly this variable, which only differed by 0.1oC showed a positive correlation. I am not sure how this could be? Was this in-situ temperature or the temperature in the cold room and if so did it vary much? Overall, I would suggest removing a number of these variables which showed very little in-situ variation and the variations were probably biologically not meaning full differences such bottom water DO, and temperature as well as depth.

Reviewer #2 (Remarks to the Author):

McTigue et al. present measurements of denitrification, anammox and DNRA from the Hanna Shoal area of the Chukchi Sea. The denitrification rates are subdivided into the fraction that is supported by NO₃ flux from the overlying water and that that is supported by within sediment nitrification. The data appear to be solid and the conclusions are logically drawn from them. To me the most significant of these conclusions are: (1) anammox and DNRA are insignificant N-cycling pathways, (2) the majority of the denitrification is derived from in situ nitrification and (3) Arctic Ocean sediments are potentially significant, but under sampled sites of global marine N₂ production. Nevertheless, the manuscript needs major modification before it is appropriate for Nature Communications. I would like to see a discussion of how doubling or tripling of the in overlying concentrations of N-species affects the rates. The manuscript also contains many unspecific words and phrases, extraneous text, confusing structure and confusing definitions.

I think the authors need to discuss the effect of increasing the overlying water concentration, especially those of nitrate. The overlying water nitrate concentration was about 5uM and they increased it by about a factor of 6 to 30uM. Nitrate pore-water profiles in the Hanna Shoal area indicate that NO₃ is exhausted within the upper 0.5-1.0 cm of the sediments (Chang and Devol, 2009). Consequently, this increase in overlying water nitrate will drive a large flux of nitrate into the sediment. Not only will the flux be increased, but also the nitrate penetration depth will be increased. Somewhere in the methods section they say their rate estimates are conservative, however I would think increasing the flux and penetration depth of nitrate into the sediment would increase rates.

I found the organization of the manuscript quite confusing. In the discussion section they start talking about D15, D14, A14 etc. before defining them. Consequently, every time I came across a new, undefined term I had to go to the end of the paper where the methods section was to figure out what they were talking about (or worse the supplemental information). I realize that it is the style of Nature publications to put the methods at the end, but some simplified definition of these terms would have been helpful. Along the same vein, what is important here is the rate of Denitrification, the amount of denitrification supported by overlying water nitrate, the rate supplied by within sediment nitrification and anammox. Why not just do the discussion of those in those terms and leave the D14, D14, A15, etc out of the main text.

And, while we are on methods, I am having a hard time understanding equation 3. But first, why in eq 2 is it p29no3 and p30? Why not p30no3? , Ok now back to eq 3, how do they get D14 from only a measurement of the things that come from adding 15NO₃. Don't they need the 14/15 ratio of the nitrate in the feed water? Potentially you can get D14 from the binomial distribution, but this involves squared terms. What am I missing here?

Finally, the manuscript contains many unspecific or incorrectly used terms, and extraneous phrases and other technical issues. I list quite a few below, but I have likely missed some.

1) In the abstract there is an "*" after the first author's name. This usually leads to a note, presumably identifying corresponding author or something, but it is just hanging there without explanation.

2) On line 23 they define denitrification as the transport of fixed nitrogen to N₂. Then on line 32 they define "canonical denitrification: as a "microbially-mediated anaerobic transformation of nitrate to N₂", which is the same as their definition of denitrification on line 23 and would seem to include anammox. This is immediately followed by anammox as a "second pathway" (the third one discussed so far), which removes fixed nitrogen. This is the same definition as "denitrification" defined on line 23. Basically there two process: heterotrophic or chemolithotrophic denitrification which involves only NO₃ or NO₂, and anammox which involve NH₄. Define them once unambiguously and stick with those definitions.

3) Line 43. "While able to occur at relatively high rates compared to nitrogen" They state this as though it was a generality but cite a mangrove sediment paper.

4) Line 45. Delete "the presence of" it's not necessary.

5) Lines 47-49. They have not defined the "system", but sediments are part of it. The paper is about sedimentary N-cycling so why sediments are not part of THE SYSTEM? I would replace "system" with "ecosystem".

6) Line 55. Dalsgaard et al. This is a pretty old paper, and more recent papers both support and contradict it. I would delete this sentence.

7) Line 77. "Crucial" is the wrong word. Substitute "successful".

8) Line 84. "the gateway sea" is this in general, or only from an North American perspective? Most of the water in the Arctic Ocean comes from the Atlantic.

9) Line 87. "Concentrations" is the subject, so "fuel" is the verb.

10) Lines 88-91. This sentence is a mess. Does "that" refer to "blooms" or "ice"? "food for a substrate for benthic...."? Why not just "food for benthic...".

11) Lines 91-92. Delete the phrase "due to its hydrodynamic advantage over other areas" because its redundant with "as the currents eddy around..".

12) Line 93. I would also delete "Yet amidst the rich ecosystem".

13) Line 101. Replace "work measuring" (dangling participle) with "measurements of".

14) Line 102. "benthos". Do you mean benthos or sediments?

15) Line 109. The term DNRA14 is undefined.

16) Line 113. "stations occupied for core collection" Just say "sampling stations" if there were no other kind. Also "(Table 1)", they need to somehow indicate that all the other things in this paragraph are also in table 1, not just the sampling stations.

17) Line 128, sentence starting with "While D15". I'm not sure what they are getting at here. If D15 is confounded, the D14 is also because it is calculated from D15 and Dtot.

18) Line 134. "Anammox (A14)", why not just say anammox and forget the A14? And why switch

to A14 for the rest of the Paragraph?

19) Reference line 168. Ref 43 actually says DNRA favored over denitrification at high loadings (last sentence of ref 43 abstract).

20) Lines 168-170. What's the point of this speculation about downslope? They have no rate data from downslope and neither do they have any OM delivery data?

21) Line 172 to 175. The sentence starting with "How ra" is all speculation. I would eliminate. Basically, I would eliminate all unsupported speculation in this paragraph and replace it with the paragraph starting on the next page.

22) Line 183. Replace "Conductive" with "favorable".

23) Line 205-206. What's the point of the sentence starting with "Without Nitrogen fixation"? It's pretty clear by now that the main N source is the Pacific inflow through Bering Strait.

24) Line 208 I would replace "facilitated" with either "fueled" or "driven".

25) Line 210. "Sediments contributing". I would say "recycling" because nearly all the N ultimately comes from the North Pacific.

26) Line 222. They don't have to "postulate", their data "show" it.

27) Line 225. "rates might be relatively high". They can do better than this. If Dn is half of denitrification, then nitrification rate must be at least half of the source.

28) Line 234. "temperature". The word they want is "temperate" Also what's the point of this paragraph? They have no data on temperature affects o arctic denitrifiers and it's obvious from the data that denitrification exists in the ocean?

29) Line 243. "those authors" No authors have been named in this paragraph. I don't think a pronoun can refer to a footnote. The same is true for "(ibid.)".

30) Line 250. "these" Which ones are these? No stations have been specified. Do they mean "Our" or some subset of their stations?

31) Line 252. Again, what does "these" refer to? What dynamic patterns?

32) Line 260. What are "station rates"? How fast they did the stations?

33) Line 261. What is potential bioturbation. Please explain briefly what this index is composed of.

34) Why say "It is possible that infaunal bioturbation was not equally important for predicting rates across the entire system"? Just say "It was the strongest correlating factor in this study."

35) Basically isn't latitude a substitute for temperature. Why not put this with the temperature discussion??

36) Line 314. "undermines" is wrong word. Use "neglects" or "underestimates" whichever they mean.

37) Line 386. "entrained" is the wrong word. Either they mean "constrained" or something else.

38) Line 404. "spanned across", One of these words is redundant.

39) Line 405. "entrained" again. Maybe they need to look up this word.

40) Line 436. I assume the samples were kept frozen until analysis at UTSMI.

41) Line 440. They say they measured $^{28}\text{N}_2$ (actually mass 28). But they never discuss it. If they measured mass 28 they should have been able to get an N_2 production in the control by just looking at the time series. Did they do this? Why mention mass 28 and not present any mass 28 data?

42) Line 569. " n_2 " should be capitalized and subscripted.

43) Line 859. "Bold values are trends" A single value is not a trend. I'm pretty sure they mean "significant correlations?"

44) Within Table 3 itself. Again, what composes "BPC" and "total abundance" of what. Also the significant correlation of SOD with Temperature (-0.94). Really a significant correlation when the total temperature range is 0.1C? Is the difference of 0.1C really a significant difference in temperature?

Reviewer #3 (Remarks to the Author):

This paper examined N_2 loss rates and its controls at 5 stations on an Arctic shelf at a single point in time. They found that canonical denitrification rates were not uniform across the 5 stations, but were always higher than anammox and DNRA. Most of the fueling nitrate for canonical denitrification was from coupled nitrification-denitrification in the sediments. The authors conducted a correlation analysis with N_2 loss rates and a bioturbation index and reported that infauna enhance denitrification and DNRA. They conclude with a global analysis of sediment denitrification and suggest that N_2 loss is at least as important in Arctic shelves as in lower latitudes.

Overall, the methods, analysis, and statistics in this paper are clear, appropriate, and technically sound. The correlation analysis of potential bioturbation and denitrification is an important finding and has not been reported for Arctic shelves to my knowledge. The conclusions are robust and valid according to what was presented in the paper. The paper was overall clearly written.

The most novel aspect of the paper is the summary analysis of N_2 loss rates, r_a , and DNRA with latitude, but I had some reservations about this, see below. The paper would be of interest to nitrogen cycle researchers. In my opinion, the paper likely belongs in a specialty journal.

I have a few concerns with some items in the paper

1) The authors should include a discussion about RKR organic matter stoichiometry and the expected contribution of anammox and canonical denitrification to total N_2 loss based on stoichiometry (i.e. see Babbin et al. 2014 Science vol 344 and ref therein). This is missing from the paper and is an important recent development in the N cycle. Anammox is expected to account for ~28% of total N_2 removal based on Redfieldian OM, and deviations from this represent interesting controls in the system

2) The authors need to be more clear about terms denitrification (= N_2 removal) vs. canonical heterotrophic denitrification. Occasionally in the discussion, " N_2 loss" would be better substituted for "denitrification".

3) I was curious about the relationship between temperature and N_2 loss. First, the authors give apparently conflicting data for the optimum temperature for heterotrophic denitrification (see lines 63 & 230. Second, in fig 4, the authors plot denitrification, r_a , and DNRA in relation to latitude. While this is a neat compilation and summary of global sediment denitrification, the more

important question for their paper (given the title) is the regulation of temperature upon denitrification rates as well as the potential need for revision of optimum temperature for heterotrophic denitrification (given no apparent relationship between latitude and rate). Also, given the paper's focus on abiotic controls, I was unclear why the authors plotted latitude instead of temperature.

4) Some key results are missing from the results section, including infauna counts and SOD. Instead, they were presented for the first time in the discussion.

5) Bioturbation has been reported to be important in other elemental cycles in the Bering Sea, see Davenport et al. DSRII 2013.

6) I did not see any mention of sensitivity or reproducibility of their mass spectrometry measurements nor geochemistry measurements. This would help readers understand the error of your measurements.

7) In lines 215-216, the authors state that studies with N₂ flux are not directly comparable to their study, but I was not satisfied with a reason. Both studies are measuring total N₂ removal processes, so further explanation is necessary. Along these same lines, Horak et al. DSRII 2013 measured N₂ flux in Bering Sea sediments, and would be a good comparison to this study.

Reviewers' comments are copied below. Authors' responses are interspersed between each comment in blue, italicized font.

Reviewer #1:

This paper documents the importance of different nitrate reduction process and N₂ fixation in shallow Arctic shelf sediments. The work is important for several reasons. First, the Arctic shelf is a large area making up nearly 20% of the continental shelf area and therefore the Arctic shelf may be very important in the overall oceanic N budget. Second is that some, but not all, previous studies have suggested that anammox may be more important in cold sediments than at lower latitudes. Additionally, this study ties in nitrogen cycling to bioturbation, something that is thought to be important but rarely examined in a quantitative way. Finally, this study has examined DNRA, something that has not been studied in the Arctic shelf previously.

The paper is well written but could be improved somewhat for clarity and there are a number of issues the authors need to comment on concerning methods.

What temperature were the experiments actually run at? The authors say in a 4°C environmental chamber. Were they at 4? The authors should make a note in the discussion that if so this was at least 5°C higher than bottom water temperature. I doubt this effected the partitioning but it possibly affected the overall rates? This needs to be mentioned and the possible impact of this on the overall extrapolation of rates considered.

We greatly appreciate Reviewer 1's substantive comments.

All incubations were maintained at 4°C for the duration of the experiment. We have noted that this temperature is warmer than the in situ bottom water temperature and that it could possibly have accelerated rates but did not likely influence partitioning (Lines 235-238).

The authors need to carefully comment on the method they used to measure anammox. There are now several methods in use to measure anammox. All have their advantages and disadvantages and there is considerable debate about the artefacts from each. In this case the authors added ¹⁵N labeled ammonium to the overlying water in a flow through system with oxic overlying water. The ¹⁵N ammonium could be directly used for anammox but it could also be nitrified and subsequently denitrified. This would cause an overestimation of the rates. Anammox could be underestimated if the labeled ammonium did not have time to reach the zone of anammox. Ammonium adsorbs to sediments so it diffuses more slowly than nitrate but this is more of a problem in short term incubation than the flow through study here.

Reviewer 1 has correctly pointed out that there are limitations of the anammox method. Our methodology allows us to calculate anammox and denitrification using the parallel ¹⁵NH₄⁺ and ¹⁵NO₃⁻ whole-core incubations, which is a unique strength of our approach. But as Reviewer 1 notes, there are assumptions involved. The limitations of our anammox methodology have been described in further detail on L501-510.

If the ^{15}N -labeled ammonium is nitrified and subsequently denitrified, then anammox would be overestimated. Without inhibiting nitrification, this cannot be tested, but we have clearly stated this assumption in the text (L501-510).

If the $^{15}\text{N-NH}_4^+$ did not reach the anammox zone, then anammox would be underestimated. We equilibrated our flow-through conditions overnight, and measured rates for four subsequent days (L443-444). We also tested if the flow-through conditions had reached steady-state equilibrium by looking for statistical differences in outflow products over the four-day incubation (L557-567).

Finally, if a very significant amount of the added ammonium was nitrified and not denitrified, anammox could be greatly underestimated because no remaining ammonium would be available. This could be checked by looking at the ammonium and nitrate concentrations coming from the cores. These are not reported but measured? Assuming the label was not largely nitrified I suspect that the reported rates are maximum rates. This should be discussed.

We did not quantify $^{15}\text{NO}_3^-$ concentrations in the outflow, so we are unable to estimate rates of nitrification of added $^{15}\text{NH}_4^+$. Our net nutrient fluxes (not reported in this manuscript) demonstrate net ammonium production/release from sediments. Given the increase in outflow ammonium concentration and appreciable porewater ammonium concentrations (Table 1), it seems a reasonable assumption that anammox bacteria were not ammonium limited. We used equations 6 and 7 in order to express anammox in terms of the ambient ^{14}N available at our study site, which is a conservative approach to reporting, for example, compared to simply the production of $^{29}\text{N}_2$ in the $^{15}\text{NH}_4^+$ treatment (p29 $_{\text{NH}_4^+}$).

Line 129 - D15 us "confounded" I agree with the authors' points but don't think it is quite the correct word. I think a better way to say it is that because of added NO_3 , D15 don't reflect in-situ rates but have been used to assess denitrification potential. Did the D15 follow SOD better than D14 or track D14 closely? Given the low bottom water NO_3 I would have expected some stimulation of direct denitrification in sites with sufficient available C.

We changed our wording to reflect Reviewer 1's recommendation, now on L130-132. D15 tracked D14 closely (now stated on L132) but was slightly better correlated with SOD than D14 due to a higher rate at H33 (Table 2). We didn't include D15 graphically as not to detract attention away from D14, which is comparable to the other N-transformation processes and reflects the in situ denitrification rate. Yes, the extra nitrate did stimulate denitrification (as demonstrated by $D_{15} > D_{14}$) but our aim was to best represent the in situ rates, so we do compare D14 to environmental parameters. Also, as noted above, the added $^{15}\text{NO}_3^-$ should not have affected partitioning of rates.

I liked the authors extrapolation to the entire shelf. They explain all of the caveats to the three methods. While there is a lot of uncertainty in the estimates it gives some order of magnitude estimate and will help future investigators to choose stations which will best be able to reduce the uncertainty.

We thank Reviewer 1 very much for this feedback.

I appreciate the effort that went into Figure 4 and supplementary table 2 but I wonder how effective it is, and it is graphically very unappealing. I think most of the points could simply be made in the text. As useful as Table S2 it does not contain other information (like SOD) that might be used to try to tease out relationships and it includes a variety of techniques (actual, potential) methods to measure some of these rates. I feel like this table and graph could be omitted here and used for a more comprehensive review paper or meta-analysis more effectively.

We agree that only the same methods should be compared to each other. When comparing actual rates, we only use rates determined by IPT reported areally. However, in relating r_a or %DNRA to our measured ratios, we surveyed techniques that use either the areal (m^{-2}), mass (g^{-1}), or volumetric (mL^{-1}) approach. The ratios should be conserved regardless of approach. This is mentioned in the manuscript text (L314-318) and now in the Figure 4 caption. We have amended Figure 4 based on feedback from other reviewers to display rates by temperature instead of latitude. We agree that a truly exhaustive literature survey should include other parameters like SOD, but our purpose was to graphically describe the relationship of nitrogen cycling rates in perennially cold environments to warmer ones. In addition, we left the accompanying table in the Supplemental Materials, as we believe it may be of interest to others studying global nitrogen cycling and fodder for comprehensive meta-analyses Reviewer 1 describes.

It would help to casual reader who firsts looks at the graphs to define BPC in the table and figure legends.

Agreed. This has been implemented in Figure and Table legends.

The correlation matrix (table 3) includes variable which did not really show any differences between stations (such as temperature). Surprisingly this variable, which only differed by 0.1oC showed a positive correlation. I am not sure how this could be? Was this in-situ temperature or the temperature in the cold room and if so did it vary much? Overall, I would suggest removing a number of these variables which showed very little in-situ variation and the variations were probably biologically not meaning full differences such bottom water DO, and temperature as well as depth.

Good catch. The significant relationship between temperature and SOD ($p=0.018$) should be ignored: a difference of 0.1 degrees does not likely drive SOD. In this case, the two stations with $T=-1.7$ had lower SOD rates than the three stations with $T=-1.6$.

We deliberated whether to include all variables tested for correlation to be most transparent to the reader, or to include only significantly correlated variables for clarity. Given Reviewer 1 & 2's recommendation, we have removed non-significant variables from the table to improve clarity, but list them in the Table legend for transparency.

Reviewer #2 (Remarks to the Author):

McTigue et al. present measurements of denitrification, anammox and DNRA from the Hanna Shoal area of the Chukchi Sea. The denitrification rates are subdivided into the fraction that is supported by NO₃ flux from the overlying water and that that is supported by within sediment nitrification. The data appear to be solid and the conclusions are logically drawn from them. To me the most significant of these conclusions are: (1) anammox and DNRA are insignificant N-cycling pathways, (2) the majority of the denitrification is derived from in situ nitrification and (3) Arctic Ocean sediments are potentially significant, but under sampled sites of global marine N₂ production. Nevertheless, the manuscript needs major modification before it is appropriate for Nature Communications. I would like to see a discussion of how doubling or tripling of the overlying concentrations of N-species affects the rates. The manuscript also contains many unspecific words and phrases, extraneous text, confusing structure and confusing definitions.

I think the authors need to discuss the effect of increasing the overlying water concentration, especially those of nitrate. The overlying water nitrate concentration was about 5uM and they increased it by about a factor of 6 to 30uM. Nitrate pore-water profiles in the Hanna Shoal area indicate that NO₃ is exhausted within the upper 0.5-1.0 cm of the sediments (Chang and Devol, 2009). Consequently, this increase in overlying water nitrate will drive a large flux of nitrate into the sediment. Not only will the flux be increased, but also the nitrate penetration depth will be increased. Somewhere in the methods section they say their rate estimates are conservative, however I would think increasing the flux and penetration depth of nitrate into the sediment would increase rates.

We are thankful for Reviewer 2's thorough suggestions.

All reviewers noted that adding extra nitrate will typically increase denitrification rates, and we agree 100%. Adding additional tracer above ambient concentrations is common practice in ¹⁵N tracer techniques (see Nielsen et al., 1992; Giblin et al. 2010, Porubsky et al. 2009). This is required for a number of reasons, including the need to add enough ¹⁵N substrate to be able to detect it in the end product pools (see Nielsen 1992 for details of the isotope pairing technique (IPT)). The flexibility of the technique allows the author to report rates in terms of ¹⁵NO₃⁻ or ¹⁴NO₃⁻ or total NO₃⁻, and the decision is different for each study. We strictly reported our rates in terms of the ambient ¹⁴N available in situ, i.e., 'D₁₄', 'A₁₄', and 'DNRA₁₄'. Increasing ¹⁵NO₃⁻ concentrations above ambient ¹⁴NO₃⁻ also ensures we are not violating an assumption of the IPT method that the ratio of ¹⁵NO₃⁻ and ¹⁴NO₃⁻ in the denitrification zone is constant. If the NO₃⁻ penetration depth is increased by additional ¹⁵NO₃⁻, the D₁₄ rate we report still removes the effect of the extra NO₃⁻. We have added a clarifying statement on L471-472 addressing this point.

Nielsen, L. P. Denitrification in sediment determined from nitrogen isotope pairing. FEMS microbiology ecology 86: 357-362 (1992).

Porubsky, W.P., N.B. Weston, S.B. Joye (2009). Benthic metabolism and the fate of dissolved inorganic nitrogen in intertidal sediments. Estuarine, Coastal, and Shelf Science 83:392-402.

Giblin, A. E., Weston, N. B., Banta, G. T., Tucker, J. & Hopkinson, C. S. (2010). The effect of salinity on nitrogen losses from an oligohaline estuarine sediment. Estuaries and Coasts 33:1054-1068.

I found the organization of the manuscript quite confusing. In the discussion section they start talking about D15, D14, A14 etc. before defining them. Consequently, every time I came across a new, undefined term I had to go to the end of the paper where the methods section was to figure out what they were talking about (or worse the supplemental information). I realize that it is the style of Nature publications to put the methods at the end, but some simplified definition of these terms would have been helpful. Along the same vein, what is important here is the rate of Denitrification, the amount of denitrification supported by overlying water nitrate, the rate supplied by within sediment nitrification and anammox. Why not just do the discussion of those in those terms and leave the D14, D14, A15, etc out of the main text.

We had not taken the structure of the Nature publications into account, and we really appreciate Reviewer 2 for pointing this out! Most readers will read the Methods last, if at all. We have added definitions to our terms in the beginning of the Results on L120-124. We have elected to leave in the terminology D_{14} , A_{14} , $DNRA_{14}$ in the results when we specifically discuss our measured rates. There are many ways to report these processes, and not all of them are intercomparable. This is a common source of frustration in the ^{15}N tracer literature, which is why we choose to be explicit. We feel it is a pertinent reminder so the reader is aware of exactly how we measured these rates.

And, while we are on methods, I am having a hard time understanding equation 3. But first, why in eq 2 is it $p_{29}\text{no}_3$ and p_{30} ? Why not $p_{30}\text{no}_3$?

We realize the terminology can be confusing, but we have used the terms used in previous studies for continuity. $p_{30}\text{NO}_3^-$ is the correct nomenclature since the $^{30}\text{N}_2$ is derived from $^{15}\text{NO}_3^-$. We omitted it to be concise. $^{29}\text{N}_2$ can be formed from the $^{15}\text{NO}_3^-$ or $^{15}\text{NH}_4^+$ treatment, depending on if denitrification or anammox is responsible for the N_2 production, so it is specified in the equations, e.g., Eq. 6 requires the $^{29}\text{N}_2$ produced from $^{15}\text{NH}_4^+$ and is different than the $^{29}\text{N}_2$ produced from $^{15}\text{NO}_3^-$. These definitions are now on L473-474, L482 and L496-497. See Risgaard-Petersen et al. (2003) for further discussion.

Risgaard-Petersen, N., Nielsen, L. P., Rysgaard, S., Dalsgaard, T. & Meyer, R. L. Application of the isotope pairing technique in sediments where anammox and denitrification coexist. Limnology and Oceanography - Methods 1: 63-73 (2003).

Ok now back to eq 3, how do they get D14 from only a measurement of the things that come from adding $^{15}\text{NO}_3$. Don't they need the 14/15 ratio of the nitrate in the feed water? Potentially you can get D14 from the binomial distribution, but this involves squared terms. What am I missing here?

Equations 2 and 3 are directly from Nielsen (1992) who developed the isotope pairing technique (IPT). The technique is used to quantify the amount of ^{14}N - N_2 produced as it would occur without

*the addition of $^{15}\text{NO}_3^-$, which is why we report D_{14} . p29/2*p30 is related to $^{14}\text{N}/^{15}\text{N}$, but the equation uses the former terms instead. See Nielsen (1992) and Risgaard-Petersen et al. (2003) for much more detailed discussion of how the equations derive D_{14} .*

Finally, the manuscript contains many unspecific or incorrectly used terms, and extraneous phrases and other technical issues. I list quite a few below, but I have likely missed some.

1) In the abstract there is an "*" after the first author's name. This usually leads to a note, presumably identifying corresponding author or something, but it is just hanging there without explanation.

This was an oversight. Yes, "" corresponds to first author and has been updated.*

2) On line 23 they define denitrification as the transport of fixed nitrogen to N_2 . Then on line 32 they define "canonical denitrification: as a "microbially-mediated anaerobic transformation of nitrate to N_2 ", which is the same as their definition of denitrification on line 23 and would seem to include anammox. This is immediately followed by anammox as a "second pathway" (the third one discussed so far), which removes fixed nitrogen. This is the same definition as "denitrification" defined on line 23. Basically there two process: heterotrophic or chemolithotrophic denitrification which involves only NO_3 or NO_2 , and anammox which involve NH_4 . Define them once unambiguously and stick with those definitions.

We thank Reviewer 2 for pointing out this ambiguity, as did Reviewer 3. We have carefully altered our wording throughout the manuscript, and particularly in the Introduction, L23-39.

3) Line 43. "While able to occur at relatively high rates compared to nitrogen" They state this as though it was a generality but cite a mangrove sediment paper.

We agree this was poorly phrased. Our verbiage has been updated to "The controls on the environmentally variable nitrogen fixation process are not well understood but appear to be regulated by P and Fe availability." We removed the reference to Alongi et al. (2000) and now only cite a recent paper by Landolfi et al. (2015) that investigates the controls on nitrogen fixation from a global perspective. Now on L42-43.

4) Line 45. Delete "the presence of" it's not necessary.

Done.

5) Lines 47-49. They have not defined the "system", but sediments are part of it. The paper is about sedimentary N-cycling so why sediments are not part of THE SYSTEM? I would replace "system" with "ecosystem".

Done. L44.

6) Line 55. Dalsgaard et al. This is a pretty old paper, and more recent papers both support and contradict it. I would delete this sentence.

The paragraph has been revised to be more succinct and include more recently published papers (Lisa et al. 2014, Babbitt et al. 2014, Devol 2015, and Plummer et al. 2015), now on L49-52.

7) Line 77. "Crucial" is the wrong word. Substitute "successful".

This sentence has been deleted during the revision process.

8) Line 84. "the gateway sea" is this in general, or only from a North American perspective? Most of the water in the Arctic Ocean comes from the Atlantic.

The more-correct term we now use is "Pacific gateway sea" since a mean northward advection of Pacific-derived water enters the Chukchi Sea via the Bering Strait. We have updated the text to reflect the change. L73.

9) Line 87. "Concentrations" is the subject, so "fuel" is the verb.

This sentence has been edited during revision. See next comment.

10) Lines 88-91. This sentence is a mess. Does "that" refer to "blooms" or "ice"? "food for a substrate for benthic...."? Why not just "food for benthic...".

The sentence now reads "High water column DIN accumulation during ice-covered periods fuels phytoplankton and sea ice algae blooms that reach the seafloor largely ungrazed, ultimately supplying organic matter for macrofaunal and microfaunal food webs." L77-79.

11) Lines 91-92. Delete the phrase "due to its hydrodynamic advantage over other areas" because it is redundant with "as the currents eddy around..".

The sentence now reads "The Hanna Shoal region is an ecological hotspot since currents eddy around the shoal and deposit organic matter to the seafloor." L79-81.

12) Line 93. I would also delete "Yet amidst the rich ecosystem".

Done.

13) Line 101. Replace "work measuring" (dangling participle) with "measurements of".

Done. L91-92.

14) Line 102. "benthos". Do you mean benthos or sediments?

We meant 'sediments'. The sentence now reads "Previous measurements of the net flux of N₂ from sediments indicate that the Chukchi Sea sediments are a sink for bioavailable nitrogen." L91-92.

15) Line 109. The term DNRA₁₄ is undefined.

Good catch. We use the term DNRA, which has been defined, instead. L96.

16) Line 113. "stations occupied for core collection" Just say "sampling stations" if there were no other kind. Also "(Table 1)", they need to somehow indicate that all the other things in this paragraph are also in table 1, not just the sampling stations.

The Results section now begins with a sentence that indicates all physio-chemical parameters are reported in Table 1. We have implemented the more-succinct term "Sampling stations..." L109.

17) Line 128, sentence starting with "While D₁₅". I'm not sure what they are getting at here. If D₁₅ is confounded, the D₁₄ is also because it is calculated from D₁₅ and D_{tot}.

The term "confounded" does not have the proper connotation, as Reviewer 1 also pointed out. We have re-phrased the sentence on L130-132. What was meant was that D₁₅ cannot be interpreted as the in situ denitrification rate because adding NO₃⁻ to a system increases the total denitrification rate. We use the isotope pairing technique (IPT; Nielsen 1992) to calculate D₁₄, which is reflective of the in situ denitrification rate. However, we can use D₁₄+D₁₅ (as in other IPT studies) as an indicator of denitrification potential, i.e., if there was additional nitrate present, could the microbial community denitrify it? We have revised the statement to "Although rates of denitrification of ¹⁵NO₃⁻ (D₁₅) are not reflective of in situ denitrification, they do show..."

18) Line 134. "Anammox (A₁₄)", why not just say anammox and forget the A₁₄? And why switch to A₁₄ for the rest of the Paragraph?

A₁₄, D₁₄, and DNRA₁₄ notation is used to indicate that these terms were defined relative to the ambient ¹⁴N, so they reflect in situ rates. It is an important distinction to remind the reader since these rates can be presented by several different terms, reflective of different measurement techniques. For example, 'anammox' could be interpreted as 'potential anammox', which is calculated using a differently than A₁₄. We have clarified this usage prior to the Results section on L120-124, as also suggested by Reviewer 1.

19) Reference line 168. Ref 43 actually says DNRA favored over denitrification at high loadings (last sentence of ref 43 abstract).

True, very high OM loads tend to favor DNRA, as observed in sediment underlying aquaculture pens, for example. In the context of this paragraph interpreting the ra values we observed, very low loadings typically favor anammox, while the productive Chukchi Shelf would favor denitrification, which aligns with the findings of Burgin and Hamilton (2007) and Hardison et al. (2015). The OM loadings of the Chukchi Sea are much lower than those speculated to favor DNRA, as is discussed in the following paragraph in the manuscript.

20) Lines 168-170. What's the point of this speculation about downslope? They have no rate data from downslope and neither do they have any OM delivery data?

We have added citations and re-worded this sentence. This idea is now substantiated by other research conducted in the region and suggests the need for further research in this area to test this hypothesis. L168-178.

21) Line 172 to 175. The sentence starting with "How ra" is all speculation. I would eliminate. Basically, I would eliminate all unsupported speculation in this paragraph and replace it with the paragraph starting on the next page.

We thank Reviewer 2 for this feedback. Since the other two reviewers did not comment on this section, we have decided to keep it in the Discussion. We feel that forming hypotheses to test in future studies using our data in combination with previous work is an important aspect to the Discussion. Skelton and Edwards (2000, British Medical Journal 320:1269-1270) provide a compelling argument for the role of speculation in the Discussion.

22) Line 183. Replace "Conductive" with "favorable".

Done. L185.

23) Line 205-206. What's the point of the sentence starting with "Without Nitrogen fixation"? It's pretty clear by now that the main N source is the Pacific inflow through Bering Strait.

Agreed. This sentence has been deleted.

24) Line 208 I would replace "facilitated" with either "fueled" or "driven".

We have replaced "facilitated" with "driven". L209.

25) Line 210. "Sediments contributing". I would say "recycling" because nearly all the N ultimately comes from the North Pacific.

True. We have used the word "recycling" instead. L211.

26) Line 222. They don't have to "postulate", their data "show" it.

True. We have used the word "demonstrate" instead of "postulate". L226.

27) Line 225. "rates might be relatively high". They can do better than this. If D_n is half of denitrification, then nitrification rate must be at least half of the source.

We have altered the text to read: "Although we did not measure it directly, sediment nitrification must be relatively active given that D_n , which hinges on nitrification to provide NO_3^- , was 58 – 92 %." L229-230.

28) Line 234. "temperature". The word they want is "temperate" Also what's the point of this paragraph? They have no data on temperature affects o arctic denitrifiers and it's obvious from the data that denitrification exists in the ocean?

Good catch! However, we have removed these paragraphs since it detracted from the main point of the manuscript.

29) Line 243. "those authors" No authors have been named in this paragraph. I don't think a pronoun can refer to a footnote. The same is true for "(ibid.)".

This is a good point for the superscripted reference style. During revision, however, we have removed this paragraph.

30) Line 250. "these" Which ones are these? No stations have been specified. Do they mean "Our" or some subset of their stations?

"these" has been changed to "our". L232.

31) Line 252. Again, what does "these" refer to? What dynamic patterns?

This sentence has been updated to "While our sampling stations spanned the northeast Chukchi Sea shelf near Hanna Shoal, many environmental parameters that have been reported to affect nitrogen cycling (e.g., temperature, salinity, depth, season) were uniform and could not control the variation in biogeochemical rates (Table 1)." L232-235.

32) Line 260. What are "station rates"? How fast they did the stations?

We agree "station rates" is ambiguous. We have replaced it with "nitrogen transformation rates". L258.

33) Line 261. What is potential bioturbation. Please explain briefly what this index is composed of.

Good point. We have added the definition on L265-267.

34) Why say "It is possible that infaunal bioturbation was not equally important for predicting rates across the entire system"? Just say "It was the strongest correlating factor in this study."

Now L387-394. We understand Reviewer 2's suggestion to make this section more succinct. However, we feel that this caveat is important since our "scale-up" exercise assumes the correlation is constant over the entire extent of the Chukchi Sea. We did re-word the sentence to be more clear in our meaning and segue into the next sentence, which shows by example of the study by Braeckman et al. (2014) that trends can be spatially-dependent.

35) Basically isn't latitude a substitute for temperature. Why not put this with the temperature discussion??

A good point also mentioned by Reviewers 1 & 3. We have revised Figure 4 to include temperature. This makes a clearer point than using latitude.

36) Line 314. "undermines" is wrong word. Use "neglects" or "underestimates" whichever they mean.

L357. We replaced "undermines" with "neglects"

37) Line 386. "entrained" is the wrong word. Either they mean "constrained" or something else.

We replaced "entrained" with "constrained". L402.

38) Line 404. "spanned across", One of these words is redundant.

"Spanned" has been deleted.

39) Line 405. "entrained" again. Maybe they need to look up this word.

We replaced "entrained" with "constrained". L421.

40) Line 436. I assume the samples were kept frozen until analysis at UTSMI.

Yes. L451-452 have been updated to clarify this assumption.

41) Line 440. They say they measured $^{28}\text{N}_2$ (actually mass 28). But they never discuss it. If they measured mass 28 they should have been able to get an N_2 production in the control by just looking at the time series. Did they do this? Why mention mass 28 and not present any mass 28 data?

$^{28}\text{N}_2$ was measured in the $^{15}\text{NO}_3^-$ cores and used to determine if simultaneous N-fixation was occurring with denitrification (see An et al. 2001). However, rates of $^{28}\text{N}_2$ production do not differentiate between denitrification and anammox, so we used methods/equations described by Risgaard-Peterson et al. 2003, which do not use $^{28}\text{N}_2$.

*An, S., Gardner, W. S. & Kana, T. M. Simultaneous measurement of denitrification and nitrogen fixation using isotope pairing with membrane inlet mass spectrometry analysis. Applied and Environmental Microbiology **67**, 1171-1178 (2001).*

42) Line 569. "n2" should be capitalized and subscripted.

This has been corrected in our bibliography software. Thank you.

43) Line 859. "Bold values are trends" A single value is not a trend. I'm pretty sure they mean "significant correlations?"

Correct. This has been updated in the text. L898-899.

44) Within Table 3 itself. Again, what composes "BPC" and "total abundance" of what.

Descriptions have been provided in the Table 3 legend. L898-906.

Also the significant correlation of SOD with Temperature (-0.94). Really a significant correlation when the total temperature range is 0.1C? Is the difference of 0.1C really a significant difference in temperature?

This was also mentioned by Reviewer 1. The significant relationship between temperature and SOD ($p=0.018$) should be ignored: a difference of 0.1 degrees does not likely drive SOD. In this case, the two stations with $T=-1.7$ had lower SOD rates than the three stations with $T=-1.6$. Based on other comments from Reviewer 1, we have altered Table 1 so temperature is no longer included. We deliberated whether to include all variables tested for correlation to be most transparent to the reader, or to include only significantly correlated variables for clarity. Given recommendations from Reviewer 1 & 2, we have removed non-significant variables from the table to improve clarity, but list them in the Table legend for transparency.

Reviewer #3 (Remarks to the Author):

This paper examined N₂ loss rates and its controls at 5 stations on an Arctic shelf at a single point in time. They found that canonical denitrification rates were not uniform across the 5 stations, but were always higher than anammox and DNRA. Most of the fueling nitrate for canonical denitrification was from coupled nitrification-denitrification in the sediments. The authors conducted a correlation analysis with N₂ loss rates and a bioturbation index and reported that infauna enhance denitrification and DNRA. They conclude with a global analysis of sediment denitrification and suggest that N₂ loss is at least as important in Arctic shelves as in lower latitudes.

Overall, the methods, analysis, and statistics in this paper are clear, appropriate, and technically sound. The correlation analysis of potential bioturbation and denitrification is an important finding and has not been reported for Arctic shelves to my knowledge. The conclusions are robust and valid according to what was presented in the paper. The paper was overall clearly written.

The most novel aspect of the paper is the summary analysis of N₂ loss rates, ra, and DNRA with latitude, but I had some reservations about this, see below. The paper would be of interest to nitrogen cycle researchers. In my opinion, the paper likely belongs in a specialty journal.

I have a few concerns with some items in the paper

1) The authors should include a discussion about RKR organic matter stoichiometry and the expected contribution of anammox and canonical denitrification to total N₂ loss based on stoichiometry (i.e. see Babbin et al. 2014 Science vol 344 and ref therein). This is missing from

the paper and is an important recent development in the N cycle. Anammox is expected to account for ~28% of total N₂ removal based on redfieldian OM, and deviations from this represent interesting controls in the system.

We are very grateful to Reviewer 3 for these insightful comments.

We now cite Babbin et al. on L51, L243, and L245. We added a discussion of why our data deviate from the expected relationship reported by Babbin et al. This is on L242-256.

2) The authors need to be more clear about terms denitrification (=N₂ removal) vs. canonical heterotrophic denitrification. Occasionally in the discussion, "N₂ loss" would be better substituted for "denitrification".

We thank Reviewers 2 and 3 for pointing this out. We have carefully reworded the Introduction (particularly L22-47) and elsewhere to specify differences between N₂ loss and heterotrophic denitrification.

3) I was curious about the relationship between temperature and N₂ loss. First, the authors give apparently conflicting data for the optimum temperature for heterotrophic denitrification (see lines 63 & 230). Second, in fig 4, the authors plot denitrification, ra, and DNRA in relation to latitude. While this is a neat compilation and summary of global sediment denitrification, the more important question for their paper (given the title) is the regulation of temperature upon denitrification rates as well as the potential need for revision of optimum temperature for heterotrophic denitrification (given no apparent relationship between latitude and rate). Also, given the paper's focus on abiotic controls, I was unclear why the authors plotted latitude instead of temperature.

We realize the optimum temperature information presented was seemingly contradictory since one report was optimal temperature for growth while the other was for respiration. Regardless, this section has been deleted from the manuscript since it detracts from the main vein of our analysis.

We agree with all 3 reviewers that temperature should be included in our analysis, and it is now presented in Figure 4 and in the updated text (L312-334). Its inclusion has enhanced the manuscript Discussion section, so we appreciate this suggestion.

4) Some key results are missing from the results section, including infauna counts and SOD. Instead, they were presented for the first time in the discussion.

Infauna abundance and biomass, which were used to calculate the community index of potential bioturbation (BPc), were collected in a parallel study. Therefore, we do not present the data in the results or describe the methodology, but rather give credit to the other study. We obtained the data from a publicly accessible database (<http://arcticstudies.org/hannahoal/index.html>) but summarized the data for the five stations we occupied in Supplemental Table 1. This explanation is on L258-267.

SOD was measured by our study. They are described in the Methods (L428-430 & L455-456) and Results (L142-145), including Table 2 and Figure 2f.

5) Bioturbation has been reported to be important in other elemental cycles in the Bering Sea, see Davenport et al. DSRII 2013.

This citation is now included on L260. We thank Reviewer 3 for pointing us to another high latitude study.

6) I did not see any mention of sensitivity or reproducibility of their mass spectrometry measurements nor geochemistry measurements. This would help readers understand the error of your measurements.

Agreed. This was an oversight in our previous draft. Coefficients of variance and verbiage about analytical replication have been added to our methods on L455-462.

7) In lines 215-216, the authors state that studies with N₂ flux are not directly comparable to their study, but I was not satisfied with a reason. Both studies are measuring total N₂ removal processes, so further explanation is necessary. Along these same lines, Horak et al. DSRII 2013 measured N₂ flux in Bering Sea sediments, and would be a good comparison to this study.

We have added some justification on L215-225 and included a citation from Ferguson and Eyre (2007), who demonstrate why a direct comparison between rates obtained using different methods are not necessarily synonymous. Importantly, we emphasize that all previous studies and the current one find the same trend that Arctic sediments are sinks for nitrogen (L220-222). We would also like to note that a second manuscript from our group is about to be submitted to DSRII that uses the N₂:Ar method, and it does directly compare our measured N₂:Ar rates at Hanna Shoal with the other Chukchi Sea studies that use this method.

We have added a citation to Horak et al. on L86.

REVIEWERS' COMMENTS:

Reviewer #1 (Remarks to the Author):

The authors have addressed my comment and significantly improved the manuscript. I believe that it is an important contribution to our understanding of N cycling in the Arctic shelf, which appears to be of global importance in the oceanic N budget. I have some minor comments about wording. The addition of the discussion of Babbín would add more to the paper I think if it were better incorporated into the discussion of the possible role of the fauna (see below).

Lines 49-51 - I think this sentence is very confusing. How about something like : Both denitrification and anammox require suboxic conditions but differences in the concentration of nitrate, nitrite, ammonium, the presence of sulfide and the quantity and quality of OM available may favor one pathway over another.

Line 75-78 - this section is improved but still a bit confusing. I think it would help to separate the sentences and connect the dots. The DIN comes in, is there also high production during the period when ice is covering the area and this production is not grazed or does the high production which occurs during the ice free period also contribute?

119-124 to someone familiar with isotope pairing this section makes perfect sense but I do worry that many are not and like the second reviewer will worry that that data is corrupted by high nitrate additions. Can this be made clearer? Also the technique is not referenced here which I think would help. I suggest something like starting with "we used the Isotope pairing technique which allows us to separate out in-situ nitrogen transformation process in spite of the addition of a tracer (refs). However the authors can't have it both ways. On line 288 they suggest that the nitrate concentration of the tracer might have an effect. This should be clarified.

Line 180 "may not have a stronghold" somewhat odd wording.

Line 220 how about direct comparisons may be "mis-leading" rather than misguided.

Line 242 - I really like the Babbín study, but for the reasons given in this paragraph, it is very hard to compare those experiments to this study. As has been said, bulk sediments largely reflect the "ashes of the fire" not the small fraction being rapidly mineralized. A more reasonable way to get at this is by examining DIC to DIN release ratios but this does not account for differences in concentration and rates that may occur with sediments. Another possibility is that animal excretion is altering the availability of DIN vs OC. I think this discussion could be shortened and better incorporated into the discussion on bioturbation.

I am looking for a way to make Fig 4 a bit more readable. It is very hard to tell the open symbols apart, especially for 4a.

Reviewer #2 (Remarks to the Author):

I have read the revised version of the manuscript as well as the responses to my (and other referees) comments and I feel the Manuscript is ready for publication. The only comment I have is that I am still not convinced that addition of $^{15}\text{NO}_3$ it does not alter the in situ ^{14}N denitrification rate. They say that "Adding $^{15}\text{NO}_3$ increases overall denitrification rates, but the isotope pairing technique differentiates between denitrification of the added tracer and the in situ $^{14}\text{NO}_3$ ". This is true and they get the ambient 14 denitrification rate. However, the ^{14}N rate may be altered by adding ^{15}N . For example, if the addition of ^{15}N increases the penetration depth of NO_3 both the ^{14}N and ^{15}N will penetrate deeper and the 14 rate will be subsequently changed from the original also. Nevertheless, this may be a nuance that not necessary to discuss in this manuscript.

Author responses are italicized.

Reviewer #1 (Remarks to the Author):

The authors have addressed my comment and significantly improved the manuscript. I believe that it is an important contribution to our understanding of N cycling in the Arctic shelf, which appears to be of global importance in the oceanic N budget. I have some minor comments about wording. The addition of the discussion of Babbins would add more to the paper I think if it were better incorporated into the discussion of the possible role of the fauna (see below).

Again, thank you for your comments. They were very helpful.

Lines 49-51 - I think this sentence is very confusing. How about something like : Both denitrification and anammox require suboxic conditions but differences in the concentration of nitrate, nitrite, ammonium, the presence of sulfide and the quantity and quality of OM available may favor one pathway over another.

We have revised this sentence according to your recommendation. Now L50-52.

Line 75-78 - this section is improved but still a bit confusing. I think it would help to separate the sentences and connect the dots. The DIN comes in, is there also high production during the period when ice is covering the area and this production is not grazed or does the high production which occurs during the ice free period also contribute?

This section has been re-worded and clarified (L75-79). Differentiating ice-free and under-ice conditions is unnecessary and confusing for the purpose of this section. It now reads "Through the Bering Strait, the Chukchi Sea receives northerly-advected deep Pacific water containing relatively high concentrations of NO_3^- , which subsequently, fuel some of the highest primary production in all of the Arctic^{22,23}. A large fraction of this primary production is deposited onto sediments, ultimately supplying food for benthic macrofaunal and microbial food webs^{24,25}."

119-124 to someone familiar with isotope pairing this section makes perfect sense but I do worry that many are not and like the second reviewer will worry that that data is corrupted by high nitrate additions. Can this be made clearer? Also the technique is not referenced here which I think would help. I suggest something like starting with "we used the Isotope pairing technique which allows us to separate out in-situ nitrogen transformation process in spite of the addition of a tracer (refs). However the authors can't have it both ways. On line 288 they suggest that the nitrate concentration of the tracer might have an effect. This should be clarified.

We have added the technique citation to this sentence. We agree that should improve clarity. We have slightly altered the wording to be as clear as possible that the reported rates are in terms of ambient (in situ) rates. (L119-123)

The wording on L288 was unclear. It has been revised so that we clearly relate high D_n proportions to bioturbators pumping dissolved oxygen to the sediments for nitrification. The

mention of NO_3^- detracts from the purpose of this section.

Line 180 "may not have a stronghold" somewhat odd wording.

This has been changed to "may not be prevalent..."

Line 220 how about direct comparisons may be "mis-leading" rather than misguided.

Agreed and changed.

Line 242 - I really like the Babbin study, but for the reasons given in this paragraph, it is very hard to compare those experiments to this study. As has been said, bulk sediments largely reflect the "ashes of the fire" not the small fraction being rapidly mineralized. A more reasonable way to get at this is by examining DIC to DIN release ratios but this does not account for differences in concentration and rates that may occur with sediments. Another possibility is that animal excretion is altering the availability of DIN vs OC. I think this discussion could be shortened and better incorporated into the discussion on bioturbation.

Another reviewer recommended we discuss the findings of Babbin et al. in terms of abiotic regulators; thus, this section was placed after the preceding paragraph on abiotic regulators of nitrogen cycling. Measuring DIC to DIN release ratios is an interesting way of comparing sediments to the findings of Babbin et al., but we unfortunately did not measure DIC. We feel we should keep this paragraph in its current place since another reviewer requested its incorporation to our discussion; however, we have added a sentence about faunal alteration of DIN and OC in the discussion to act as a segue to the bioturbation sections. (L257-260)

I am looking for a way to make Fig 4 a bit more readable. It is very hard to tell the open symbols apart, especially for 4a.

We added a color fill for the symbols in Figure 4a and 4b. This does improve clarity. We were careful to choose a colorblind sensitive palette.

Reviewer #2 (Remarks to the Author):

I have read the revised version of the manuscript as well as the responses to my (and other referees) comments and I feel the Manuscript is ready for publication. The only comment I have is that I am still not convinced that addition of $^{15}\text{NO}_3$ it does not alter the in situ ^{14}N denitrification rate. They say that "Adding $^{15}\text{NO}_3$ increases overall denitrification rates, but the isotope pairing technique differentiates between denitrification of the added tracer and the in situ $^{14}\text{NO}_3$ ". This is true and they get the ambient ^{14}N denitrification rate. However, the ^{14}N rate may be altered by adding ^{15}N . For example, if the addition of ^{15}N increases the penetration depth of NO_3 both the ^{14}N and ^{15}N will penetrate deeper and the 14 rate will be subsequently changed from the original also. Nevertheless, this may be a nuance that not necessary to discuss in this manuscript.

Again, thank you for your previous comments in enhancing the clarity and scope of the manuscript. One of the central assumptions about the IPT is that ambient denitrification is not affected by the label addition and denitrification of water column nitrate increases linearly with concentration. You are correct in pointing out that ambient denitrification could be affected by the label addition if nitrate penetrates into zones where denitrification was not taking place. Ideally, demonstrating that this assumption is being met is tested by running the measurements at more than one nitrate concentration; however, logistically for this project, that was impossible. Practically, in our experience with IPT incubations, this assumption is always met, so we are not concerned about this possibility. However, we will include a note about this assumption in the Methods (L474-479)

Responses to Editor's comments:

1. We have altered the title so it does not contain punctuation and is within the word limit.
2. The abstract has been re-written so that it follows the format of background, a sentence that begins "Here we present...", and finally major results and conclusions. The abstract falls within the word count limit.
3. Reference to the study site has been removed from L73.
4. A statement preceding the technical details of the Methods has been added that states all samples were transported to the University of Texas Marine Science Institute (UTMSI) for analyses. (L449-450)
5. In all equations in the manuscript and supplementary information, we have used the correct mathematical format as we interpreted from the Nature style format. For example, the multiplication sign has been changed from '*' to 'x'.
6. All multipanel figures have a, b, c, etc. noted in a corner of the panel.
7. All error bars have been defined as either s.d. or s.e.m. in the figure captions.